# Green Silver Nanoparticles: An Antibacterial Mechanism

**DOI:** 10.3390/antibiotics14010005

**Published:** 2024-12-25

**Authors:** Ekaterina O. Mikhailova

**Affiliations:** Institute of Innovation Management, Kazan National Research Technological University, K. Marx Street 68, 420015 Kazan, Russia; katyushka.glukhova@gmail.com

**Keywords:** silver nanoparticles, AgNPs, green synthesis, antibacterial activity, anti-biofilm activity, anti-“quorum sensing” activity

## Abstract

Silver nanoparticles (AgNPs) are a promising tool in the fight against pathogenic microorganisms. “Green” nanoparticles are especially valuable due to their environmental friendliness and lower energy consumption during production, as well as their ability to minimize the number of toxic by-products. This review focuses on the features of AgNP synthesis using living organisms (bacteria, fungi, plants) and the involvement of various biological compounds in this process. The mechanism of antibacterial activity is also discussed in detail with special attention given to anti-biofilm and anti-quorum sensing activities. The toxicity of silver nanoparticles is considered in light of their further biomedical applications.

## 1. Introduction

Since its inception, human civilization has been continuously fighting against pathogenic microorganisms. The search for means to combat various human diseases led to the accumulation of a huge knowledge reservoir, first within the framework of traditional medicine and then drug therapy in the modern world. In 1929, Alexander Fleming discovered penicillin, an antibiotic, which marked a new era in treating ancient diseases like gangrene and tuberculosis. This discovery revolutionized the treatment of these diseases and saved countless lives. Further discoveries of other antibiotics made it possible to prevent and cure inflammatory processes caused by bacterial microbiota. However, this development also had a downside: the widespread and sometimes unregulated use of antibiotics has led to the emergence of resistant strains of bacteria, creating a new challenge for medicine. According to the World Health Organization, antibiotic resistance is making it increasingly difficult to treat infections such as intestinal infections, pneumonia, and sepsis. In addition, the burden on the health care system and its maintenance costs are seriously growing. Alternative methods of infection treatment are essential, both in terms of addressing these challenges and developing new, highly effective medications. There are many alternatives to antibiotics for the treatment of specific diseases, including therapy with bacteriophages (Agriphage, Omnilytics Ltd., Sandy, UT, USA; Listex, Micreos Ltd., Baar, Switzerland), predatory bacteria, bacteriocins (BioSafe™, bacteriocin: Nisin A; MicroGARD^®^, mixture of different bacteriocins), antimicrobial proteins, plant-derived antimicrobial substances, probiotics, and competitive destruction of pathogens. However, none of these methods has yet demonstrated effectiveness comparable to antibiotic treatment. The advantage of these approaches is that the treatment is aimed only at the pathogenic bacterium and not at other members of the commensal, beneficial microbial communities of the host. This is an important difference from most antibiotics, which generally have collateral effects on commensal bacteria in addition to the pathogenic target.

In this regard, the nanotechnology approach is gaining the most popularity, with the trend set at the end of the last century. Metal nanoparticles play a special role in it. The physical and chemical methods initially revealed many side effects, such as the toxicity of substances used in the synthesis process and their potential harm to human health and the environment. This prompted the researchers to switch to “green” nanoparticle synthesis. Lower toxicity, high stability, remarkable physicochemical properties, as well as environmental friendliness, biocompatibility and economic efficiency of metal bionanoparticles have made them real “stars” in the nanotechnology field. Eco-friendly synthesis using bacteria, fungi, algae and plants has opened up a new, amazing world of valuable particle properties in the potential medical application sphere. The discovery of the antibacterial properties of metal nanoparticles has brought their popularity to a whole new level among researchers. Green silver nanoparticles are one of the most popular objects in this regard. Mankind has been familiar with silver since ancient times—the Ancient East, Ancient Greece and Roman civilizations were engaged in silver mining. The silver antiseptic properties were known to most nations, predetermining its role in various cultural traditions. For example, in the second century AD, so-called “holy water” was known to not spoil for months. The process study in the early 20th century showed that the antibacterial properties of “holy water” are associated with the Ag^+^ presence, and it was elemental silver that had the strongest bactericidal effect [1]. The identification of antibacterial properties of biosynthesized silver nanoparticles (AgNPs) provoked a real boom in their production and research. The number of publications devoted to this problem is steadily growing. Moreover, there are already attempts to statistically analyze the publications themselves and identify trends in the antibacterial activity of AgNPs [2]. Various methods are actively applied to characterize silver nanoparticles: the shape and size of synthesized “green” AgNPs are defined by Scanning Electron Microscopy (SEM) and Transmission Electron Microscopy (TEM), UV-Vis spectrophotometry, Dynamic Light Scattering (DLS) to estimate the physical properties, and Fourier Transform Infrared Spectroscopy Analysis (FTIR), which makes it possible to characterize biomolecules involved in the silver ions reduction and nanoparticle stabilization. In addition, the X-ray diffraction is used to calculate the crystalline size of AgNPs [3]. The study of parameters such as the size, shape, and stability of AgNPs is extremely important because they largely determine the antibacterial properties of nanoparticles. “Green” AgNPs are defined as a nanomaterial with all its dimensions in the range of 1–100 nm, most often represented by spherical shapes. These have shown greater capacity and a higher surface (area-to-volume ratio) compared to silver in its bulk form. At the nanoscale, this material exhibits specific electrical, optical, and catalytic properties. High stability and evidence of antibacterial activity focused the attention of researchers and industries on this nanomaterial.

Despite the fact that the action mechanism of green AgNPs on bacterial cells has not been fully studied, a huge number of research articles are devoted to this problem now. The various factors contributing to successful biosynthesis, the biological compounds involved in the process, and their effect on the antibacterial properties of AgNPs are the most significant tasks for further AgNP applications. The assessment of metabolites, such as proteins and enzymes, synthesized at bio-factories, is crucial because they can enhance the antibacterial effect of AgNPs. The green agenda, aimed at developing eco-friendly and safe “antibiotics”, cannot be ignored either. This review focuses on the antibacterial properties of AgNPs and their impact on bacterial cells.

## 2. The Mechanism of Synthesis

The continuing interest in the biological methods for producing AgNP production is not a coincidence. Biological matrices for the synthesis of AgNPs are as diverse as the variety of biological living organisms—bacteria, fungi, algae, and plants—by which they can be produced. The factors favoring the biological agent choice for synthesis include low toxicity, energy efficiency, high productivity, and the availability of agents necessary for synthesis. This green nanosynthesis is an environmentally friendly, biocompatible method aimed at producing AgNPs with potentially useful properties for medical applications.

The scheme for obtaining AgNPs using “bio-factories” involves the interaction of silver nitrate (AgNO_3_) with biological extract (bacterial, fungal, or plant) compounds. Chemical and physical methods were traditionally applied for the synthesis of AgNPs. However, their use is accompanied by several drawbacks. The physical methods (for example, a laser irradiation method, laser pyrolysis, electrospraying, and laser ablation) require expensive equipment and high energy consumption. On the other hand, the main disadvantages of the chemical methods are that they are supposed to use highly toxic reagents (for example, sodium borohydride, formaldehyde, and methoxypolyethylene glycol), environmental pollution, carcinogenic solvents, and contamination of precursors. An important difference between physical and chemical methods is the direct biocomponent participation (proteins, enzymes, various phytocompounds (flavonoids, terpenoids, tannins, etc.)) in nanoparticle reduction and stabilization, which will be discussed later. Nanoparticles are formed in three stages: first, Ag^+^ is reduced to Ag^0^ using biological catalysts in a “factory” (organisms such as bacteria, fungi, lichens, algae, and higher plants) of synthesis; second, colloidal silver particles are formed through agglomeration of oligomeric clusters and stabilization; and third, formation of AgNPs [3] (Figure 1). Biomolecules donate electrons from functional groups like hydroxyl (-OH) and carboxyl (-COOH) to Ag^+^ ions, facilitating their reduction to elemental silver (Ag^0^) [3]. With the formation of particles, the surface plasmon resonance (SPR) increases with a gradual to sharp increase in the dark color of the medium. The color change of the resulting solution (from yellow to brown) indicates that the nanoparticles have formed [3]. Reduced silver atoms then begin to form aggregates that can be limited in size if a capping agent is present in the reaction medium; otherwise, smaller aggregates continue to grow and form larger aggregates. Capping agents interact with Ag^0^ using van der Waals forces, electrostatic interactions, or covalent bond formation between the coating substances and Ag^0^, resulting in AgNPs. The variation in the size, shape, and properties of accumulated NPs is observed due to the variation in stabilizing and reducing potential of biomolecules present in the plant.

Notwithstanding the incredible number of publications devoted to AgNP synthesis using various groups of organisms such as bacteria, fungi, lichens, algae, and higher plants, the synthesis mechanism still remains not fully investigated. The key parameters affecting nanoparticle synthesis are the pH of the solution, the extract and salt concentrations, temperature, and incubation time. However, these parameters depend on the biological sources used to synthesize the AgNPs.

### 2.1. By Bacteria

Bacteria are one of the most promising sources of AgNP production. The process can be realized both intracellularly and extracellularly according to the location of AgNP production using both Gram-positive and Gram-negative bacteria [3].

The extracellular synthesis method is preferable due to the easy and simpler purification steps (extracting the AgNPs from the solution). In contrast, the intracellular NPs synthesis method is challenging and expensive due to the involvement of additional separation and purification processes (for instance, ultrasound treatment or reactions with suitable detergents) [3]. The extracellular process can be carried out using biomass, culture supernatant, or cell-free extract. In this case, two scenarios are possible: bacterial bio-compounds released into the external environment contribute to the conversion of silver ions into AgNPs and/or nanoparticles formed inside cells are excreted outside. Such AgNPs can have spherical, disk, cuboid, hexagonal and triangular shapes. Extracellular synthesis is indicated for Gram-positive bacteria such as *Kocuria rhizophila* [4], *Planomicrobium* sp. [5], *Exiguobacterium aurantiacumm* [6], *Lactobacillus brevis* [7], *Bacillus* sp. [8], *Enterococcus* sp. [9], and Gram-negative bacteria such as *Alcaligenes faecalis* [10], and *P. aeruginosa* [11].

Nevertheless, the nanoparticle synthesis by microorganisms was also shown to occur intracellularly. Intracellular synthesis of nanoparticles includes the transportation of ions and molecules into the bacterial cells in the presence of enzymes. For example, the intracellular nature of AgNP synthesis was established for Enterobacter cloacae by El-Baghdady et al. [12]. This mechanism was also shown for Gram-positive bacteria *Corynebacterium* sp., as well as for various bacteria of the genus *Streptomyces* [3]. There are also studies demonstrating the microorganism’s ability to conduct the process both inside and outside of cells. For instance, AgNPs were biosynthesized using the supernatant and the intracellular extract of *Cupriavidus necator*, *Bacillus megaterium*, and *Bacillus subtilis* [13].

Enzymes are a fundamental factor in the bacterial synthesis of AgNPs. The main enzymes involved are nitrate reductases, which are responsible for the bioreduction of silver ions into AgNPs [7]. Nicotinamide adenine dinucleotide (NAD) and NADH-dependent enzymes are the basis for the biogenic synthesis of these nanomaterials. The reduction process is induced by electron transfer from NADH using NADH-dependent reductase as an electron transporter. Nitrate ions from the AgNO_3_ salt induce nitrate reductase. The enzyme receives electrons from NADH, and oxidizes them to NAD^+^, and then undergoes oxidation, reducing silver ions into the AgNPs (Figure 2). For example, lactobacilli are able to synthesize nitrate reductases at pH values above 6, initiating the biological conversion of Ag^+^ to Ag^0^ and the formation of AgNPs [3]. Information about respiratory and periplasmic nitrate reductases [7] is also available in the literature. In some experiments, it was found that proteins and sugars in the cell wall, where the reduction process occurs, can participate in the silver ion uptake. Another plausible mechanism for nanoparticle synthesis is based on the fact that certain bacteria generate the trans-membrane proton gradient, which is broken down by the active symport of Na^+^ ions together with Ag^+^ ions from the extracellular environment. Several silver-binding membrane proteins attract silver ions and, by deriving energy from ATP hydrolysis, result in the uptake of silver ions inside the cells and initiate the synthesis of AgNPs. It is important to note that not only nitrate reductase but also other enzyme classes are capable of reducing silver ions: extracellular keratinase *Bacillus safensis* plays a crucial role in AgNP biosynthesis [3]. The participation of periplasmic c-type cytochrome (MacA) and outer membrane c-type cytochrome (OmcF) in surface reduction in Ag^+^ to Ag^0^ was shown [3]. Chung et al. identified 17 putative genes involved in AgNP synthesis based on the gene analysis of the complete genomic sequence of the *Aggregatimonas sangjinii* F202Z8T strain [14]. These genes have potential functions for inducting the reducing factors (alkaline phosphatase, etc.) that stimulate the reduction in Ag^+^ to Ag^0^ (nitrate reductase), components of the electron transport chain (cytochrome C-type, etc.), indirectly supporting the reduction conditions, and regulating the enzyme activity (glutathione synthase, etc.) [15].

It is known that proteins can interact with nanoparticles either through free amino groups or through cysteine residues in proteins, as well as through the electrostatic interaction of negatively charged carboxylate groups in enzymes. In addition, it is believed that the presence of a carboxylate group on the bacterial cell surfaces, which causes its mostly negative charge, provides an electrostatic interaction between this group and positively charged silver ions to help capture silver ions [16]. Some amino acids, such as arginine, aspartic acid, cysteine, glutamic acid, lysine and methionine, are also implicated in the reduction of silver ions or silver nanocrystals. These act as catalysts, producing a hydroxyl ion that reacts with reducing agents such as aldehyde. It is supposed that amide linkages and carboxylate groups in proteins from *Enterococcus* sp. may bind to silver ions and convert them into nanoparticles [9]. *Planomicrobium* spp. proteins were found to be involved in the synthesis of AgNPs through FTIR analysis [5].

Interestingly, a higher protein content was observed during intracellular synthesis [16]. Moreover, the Zeta potential values of extracellular AgNPs synthesized by *Cupriavidus necator*, *B. megaterium*, and *B. subtilis* are characterized as less stable compared to intracellular nanoparticles [13]. These differences may be due to different biomolecules present in intracellular extracts, such as carbohydrates, proteins, and lipids. In the intracellular extract, the presence of DNA, c-type cytochromes, peptides, and cellular enzymes such as nitrate reductase and reducing cofactors can contribute to the synthesis and stabilization of AgNPs, preventing their aggregation and providing greater stability compared to extracellularly synthesized AgNPs. In addition, certain bacterial enzymes (silver-binding proteins, metalloproteins, and reductases) can perform intracellular synthesis of AgNPs [17].

Data on bacterial AgNPs are presented in Table 1. Obviously, nanoparticles have different sizes, but they are most often spherical in shape. The Zeta potential value indicates the surface charge potential, which is directly related to the stability of nanoparticles in suspension. The literature data indicate that Zeta potential values higher than +30 mV or lower than −30 mV indicate that nanoparticles are very stable in the dispersion solvent. AgNPs synthesized using the intracellular extracts of *B. subtilis* and *B. megaterium* were found to be stable, with Zeta potential values of −34.1 mV and −33.9 mV [13].

The cultivation conditions affect the metabolic activity of microorganisms for efficient AgNP production. Most researchers agree that the main role belongs to the pH medium, and alkaline conditions for optimal AgNP synthesis were indicated. For example, in the case of *Exiguobacterium aurantiacum*, *Escherichia coli*, and *Brevundimonas diminuta*, the AgNPs begin forming at pH 5–6, but eventually enlarge with increasing pH [9]. Similar data were obtained for *P. aeruginosa*, *E. coli*, *Acinetobacter baumannii*, and *Staphylococcus aureus* [11]. Apparently, the bacterial synthesis of AgNP was facilitated by an alkaline pH, while in acidic conditions the particles tend to aggregate and become unstable. Moreover, high pH values influenced the nanoparticle shape. At high alkaline pH, spherical and rod-shaped AgNPs were created due to a high reaction rate, while triangular or polygonal AgNPs were synthesized at lower pH due to the low reaction rates [11]. It is also likely that high pH values can increase the activity of nitrate reductase catalyzing the AgNP synthesis [17]. Moreover, the size of *B. cereus*-AgNPs decreased with increased pH, in alkaline pH, monodispersed, spherical nanoparticles formed with an increased amount. At high pH, the reaction rate was upgraded, resulting in the nucleation and growth of small-sized AgNPs [18].

The temperature is also important. The optimum temperature for AgNP synthesis by *P. aeruginosa*, *E. coli*, *A. baumannii*, and *S. aureus* was in the range from 30 to 37 °C [11], while for *K. rhizophila* it was 54.07 °C [4], and for *B. cereus* it was 48.5 °C [19], and for *Klebsiella pneumonia*—90 °C [20]. Nanoparticle size is believed to increase at low reaction temperatures and decrease with increasing temperatures.

The concentration of AgNO_3_ can also affect the size and morphology of AgNPs. At higher concentrations of AgNO_3_, the reduction rate of silver ions into AgNPs increases, while high concentrations of silver salt lead to the formation of larger aggregated nanoparticles. AgNP biosynthesis was optimal at 0.4 g/L AgNO_3_ in *P. aeruginosa* [11]. The culture medium composition is important for the most efficient biosynthesis of AgNPs. Luria-Bertani media containing peptone, NaCl and yeast extract was found to be optimal for the *E. coli*-mediated synthesis. The authors suggest that, since cells are under stress during the stationary phase, this can cause the production and secretion of small and diffusive compounds, such as exopolysaccharides, enzymes, and proteins, that are able to interact with insoluble metals and cause their reduction.

### 2.2. By Fungi

The fungal mechanism of AgNP synthesis involves the proteins that can bind to nanoparticles either through free amine groups or cysteine residues in the proteins and through the electrostatic attraction of negatively charged carboxylate groups in enzymes present in the cell wall of mycelia [21]. Additionally, some extracellular metabolites produced by fungi, such as anthraquinones, amino acids, NADH, polysaccharides, and nitrate reductase, may be involved in the reduction, capping, and stabilization of nanoparticles to prevent their agglomeration. For example, the protein participation with different molecular weights was assumed in the reduction and stabilization of nanoparticles for the AgNP synthesis by the white-rot fungus, *Stereum hirsutum* [22]. The participation of biocompounds identified as *β*-1,3-glucanase and chitinase enzymes was determined for AgNPs synthesized by *Trichoderma harzianum* [23]. The synthesis process, as a rule, is extracellular. A probable mechanism for metal ion reduction uses shuttle quinone and the enzyme nitrate reductase. The participation of nitrate reductase was discovered in the AgNP mycosynthesis by *Cladosporium cladosporioides*. The authors suggest that small molecules, such as phenolic compounds, can be involved in this process of silver salt bio-reduction [24]. The production of “naphthoquinones” and “anthraquinones” synthesized extracellularly was reported to be capable of reducing metal ions to nanoparticles. Fungal nanoparticles vary in size: for AgNPs obtained from *Penicillium oxalicum,* they are 13–23 nm [25], and for *Fusarium solani*—7.65 to 18.89 nm [26]. The most common shape is spherical, although other forms are also found, for example, cubic for *Agaricus bisporus*-AgNPs [27].

The optimal conditions for the synthesis of fungal nanoparticles are high pH values—12 values for *Stereum hirsutum* [22], and 9–12 for *Phoma sorghina*, while in acidic pH, aggregates were observed [28]. It was hypothesized that pH may induce ionization through the transfer of electrons at an alkaline pH of 8, with different metabolites present in the solution along with AgNO_3_ salt, which resulted in the formation of AgNPs in *Aspergillus terreus* [29]. An even lower, neutral pH was optimum for the effective AgNP synthesis by *Aspergillus flavipes* at pH 7 [30]. Probably, the pH value is fundamental for the compounds responsible for biosynthesis, and the optimal pH value for their activity is optimal for AgNP biosynthesis. At certain pH levels, biomolecules can be inactivated, which has an important effect on biosynthesis.

Temperature also influences AgNP mycosynthesis. Thus, for *A. terreus*, synthesis occurs at 35 °C, and a significant slowdown in synthesis at temperatures above 45 °C may probably be due to loss of protein activity and denaturation [29]. *P. sorghina* has better mycosynthesis at 90 °C, but AgNPs form aggregates within a very short time [28].

An interesting fact is that lighting also can affect the mycosynthesis of nanoparticles. The mycosynthesis of AgNPs by the *Phoma* sp. is optimal at blue-light and minimum at red light. It has long been known that fungal kingdom members, which are heterotrophic, can respond to wavelengths of light from UV-Vis to far-red and blue light sensors. Blue light is the most effective type of light in fungal photomorphogenesis. The effect of blue light is often either stimulatory or inhibitory to the developmental transition. Proteins function as photoreceptors and contain a signature motif for flavin binding, and between a conserved cysteine residue in the Light-oxygen-voltage-sensing domain and the flavin (FAD or FMN), a blue light-dependent adduct is formed, or the protein having flavin binding motifs can be photosensitized by blue light [28]. In addition, *P. sorghina* showed an increase in the mycosynthesis of AgNP with an increase in the substrate concentration up to 0.9 mM, with the optimum mycosynthesis at 0.9 mM silver nitrate concentration, and the rate of mycosynthesis of AgNP by *P. sorghina*, increased with an increase in fungal filtrate concentration [28]. The fabrication of silver nanorods by *P. sorghina* might be due to the secretion of anthraquinone derivatives. The authors have proposed a three-step mechanism for the mycosynthesis of silver nanorods. The first step is nucleation, which involves the role of proteins acting as capping agents and anthraquinone derivatives to initiate silver nanorod fabrication. The second step is elongation, in which anthraquinone derivative acts as an electron shuttle, which takes up the electron donated by inorganic nitrate and photosensitized aromatic compounds from fungal filtrate by transferring them to silver ions and leading to reducing them to form AgNP (Ag^0^). The third and final step is the termination of the silver nanorod synthesis process: the process will be terminated once the anthraquinone molecule involved in synthesis is either recruited by another nucleation center for elongation or till the distance an anthraquinone can act as an electron shuttle [28].

### 2.3. By Algae

Algae are a fascinating object for the production of nanoparticles due to their abundance of biologically active metabolites with various biological properties. Specifically, tannins, which have antibacterial properties, are abundant in algae. However, despite this potential, there is limited information available about the “factories” that produce AgNPs from algae. The available data suggest that spherical nanoparticles are the most common shape, and their size varies depending on the algae species used, ranging from 10 to 150 nm.

The bioreduction of a metal ion into nanoparticles occurs on the surface of an algal cell extracellularly, whereas, enzymatic reduction happens inside the cell walls and cell membranes by the intracellular mechanism [31]. The carbonyl group and the protein peptides can act as prospective participants in AgNP biosynthesis, and silver reduction can also be induced by nitrate reductase. The process is initiated by electron transfer from NADH to NAD^+^, catalyzed by NADH-dependent reductase as an electron transporter, which reduces Ag^+^ ions to AgNPs. Alginic acid and proteins in the *Laminaria japonica* extract were involved in reducing AgNO_3_ and stabilizing AgNPs [32]. Polysaccharides and proteins acted as reductants, while polysaccharides/fatty acids acted as stabilizers during the synthesis of *Planophila laetevirens* AgNPs [33]. Other biomolecules reported to be associated with the algal synthesis include polysaccharide for *Chlorella vulgaris* and C-phycocyanin for *Spirulina* sp. [34].

During AgNP synthesis, algal biomolecules act as both a reductant and a stabilizer. AgNPs synthesized at pH 9 demonstrated a high SPR indicating that the reducing agents, capping and stabilizing agents present in the *Clorella minutissima* extract were highly effective at pH 9 [35]. A lower pH can disrupt these natural polymers, thus destabilizing AgNPs. A lower pH of the reaction mixture facilitates large-sized Ag-NPs which degrade easily, while a higher pH leads to smaller AgNPs faster, which are also highly dispersed throughout the medium. At higher pH, the ionization of algal functional groups and reduction rate are higher [35]. The Zeta potential values and the corresponding behavior of AgNPs indicate the fundamental role of alkaline pH conditions in the synthesis and stabilization of AgNPs. The strongly negative surface charge of the synthesized AgNPs correlated with the high degree of stability of nanoparticles from *L. japonica* [32]. The algal nanoparticle synthesis is influenced by temperature. For example, excellent *L. japonica*-mediated nanoparticle production of AgNPs was at higher temperatures—90 °C and 120 °C [32]. The optimal temperature for AgNP synthesis from *Planophila laetevirens* was observed to be at 80 °C. Increasing the incubation time at 80 °C resulted in smaller NPs [33]. A possible explanation is that an increase in temperature to a certain degree activates the reductants (biomolecules) during synthesis, while raising the temperature above the optimum level could result in biomolecule degradation, agglomeration of nanoparticles or stopping the reduction reaction [33].

### 2.4. By Plants

Plants are the main treasury for AgNP production. The manifold of biological compounds is potentially capable of participating in AgNP synthesis, making plants an inexhaustible resource for receiving nanoparticles with diverse properties. Plant biosynthesis is simple and involves the interaction of AgNO_3_ with the biomolecules of plant extracts. Nanoparticle formation transpires in three stages: the ion reduction reaction leads to cluster formation, and then stimulates nanoparticle growth. Each step has unique characteristics depending on the reducing agent, its concentration, AgNO_3_, temperature, and pH. Data on the morphology and size of plant-mediated AgNPs are presented in Table 2.

The presence of hydroxyl groups (-OH) in plant biomolecules, such as amino acids, proteins, alkaloids, flavonoids, polyphenols, enzymes, tannins, carbohydrates and saponins is associated with a reduction in silver ions (Ag^+^) to Ag^0^ and stabilization [36]. Phenolic compounds, such as flavonoids and tannins can play the role of reducing agents. These phenolic compounds also have been suggested to possess chelating ability due to the highly nucleophilic nature of the aromatic rings. Functional groups like -OH and carboxyl (-COOH) in these biomolecules play a crucial role in the reduction process. Upon dissolution in water, AgNO_3_ dissociates into Ag^+^ cations and NO_3_^⁻^ anions. The negatively charged O⁻ in phenols or COO⁻ in organic acids establishes electrostatic interactions with the positively charged Ag^+^ ions. This interaction enables the donation of electrons, reducing Ag^+^ to Ag^0^ and forming AgNPs [37].

Flavonoids contain various functional groups, that have an enhanced ability to reduce metal ions by producing reactive hydrogen atoms through tautomeric transitions. The synthesis by *Euphorbia serpens* involves the participation of the keto form of the flavonoid, which transforms into the enol form with the release of hydrogen (reactive); however, due to the presence of two -OH groups on the same carbon, the enol form was unstable and converted back into the keto form. Therefore, the released reactive hydrogen converts Ag^+^ into Ag^0^, which combines with each other to form AgNPs [38].

Terpenoids can help in reducing metal ions by oxidizing aldehydic groups in molecules to carboxylic acids. Lan Anh Thi Nguyen et al. showed that the presence of β-amyrin and lupeol with the –OH group in their ring may undergo oxidation and get converted to quinone forms of β-amyrone and lupenone, respectively [39]. The synthesized AgNPs are stabilized by unshared electron pairs and *p*-electrons of the lupenone and β-amiron quinone structures, as well as other organic compounds in an aqueous extract of *Callisia fragrans* leaves. At low solution pH values, oxidation reactions of β-amirin and lupeol proceed unfavorably, therefore, the ability to reduce Ag^+^ to Ag^0^ decreases. At the same time, with an increase in pH solution, the equilibrium of the β-amirin and lupeol oxidation reaction shifts to the right, enhancing the rate of the Ag^+^ → Ag reduction reaction and the amount of the synthesized nanoparticles [39]. It is assumed that the chemical components present in the aqueous extract of *Salvia hispanica* (Chia) seeds first form a chelate with the Ag^+^, and later reduce it to Ag^0^ by the oxidation of polysaccharides in the extract [40]. Ahluwalia et al. suggested that xanthones in *Swertia paniculata* extract can act as powerful reducing agents, and similarly, the carboxylate group in proteins can be a surfactant, leading to stabilization during synthesis [41]. The presence of proteins and enzymes in the *Terminalia arjuna* bark extract favors the reduction in Ag^+^ ions to pure AgNPs and their stabilization [42]. Further, the peptide bonds in the proteins and enzymes in the leaf extract undergo degradation to form smaller peptides. The carboxylate groups present in the peptides possess a higher affinity to play a surfactant role, forming a protein corona layer on AgNPs, which stabilizes the nanoparticles [42].

The mechanism of silver nanoparticle synthesis can be associated with the presence of alkaloids atropine, apoatropine, hyoscyamine, and belladonna, where surface silver ions are retained and subsequently reduced by other phytochemicals, resulting in the materialization of silver nuclei. These formed silver nuclei accumulate and increase in size, adduce to the AgNP formation [43]. Catechins are polyphenolic compounds containing hydroxyl groups and methyl groups in benzene rings capable of catalizing AgNP synthesis in *Camellia sinesis* [44]. Gallic acid in *Eucalyptus camaldulensis*, arjunolic acid, arjungenin, arjunetin, and luteolin in *T. arjuna* may be reducing agents for silver ions [42]. The Ag^+^ ion reacts with epicatechin during *Litchi chinensis*-mediated synthesis to form an intermediate product. The Ag–epicatechin complex, further oxidized to form quinone, hydrogen atom, and electrons. The electron reduces the Ag^+^ ion to the Ag atom [45]. In an alkaline medium, the hydroxyl group of chlorogenic acid releases two H^+^ accompanied by two electrons to reduce two Ag^+^ to become Ag in *Ferula persica* [46]. Polyphenols present in plant extracts of matico (*Buddleja globosa*), such as verbascoside and/or luteolin may be related to the chemical reduction in silver ions Ag^2+^ to Ag^0^, allowing the nucleation and growth of spherical AgNPs [47].

In addition to synthesis, various biomolecules contribute to nanoparticle stabilization. Enzymes, glycosides, and saponins can help stabilize the nanoparticles synthesized by *Linum usitatissimum* [48]. The metal salts added to herbal extracts bind silver ions to water-soluble chemicals via -OH and -COOH groups. Most researchers agree that these phytochemicals initiate AgNP formation and prevent their further aggregation by coating the particles and providing capping agents enriched with hydroxyl groups and electrons [49]. Indole-3-acetic acid, L-valine, triethyl citrate, quercetin-3-O-*β*-D-glucopyranoside act as reducing and capping agents in the biosynthesis by *Ricinus communis* [50]. The protein carboxylate group has a high affinity for acting as a surfactant, forming a protein layer on nanoparticles and ultimately stabilizing AgNPs [51]. Proteins can bind to AgNPs via free amino groups or cysteine residues, thereby stabilizing AgNPs formed due to surface-bound proteins in *Protium serratum* [52]. The bioactive compounds, such as naringin, naringenin, and hesperidin containing -OH functional groups in the extract, might be responsible for the synthesis and stabilization of *Citrus maxima*-mediated AgNPs [53]. The complexation of polyphenols with metallic silver and this bond with biomolecules is responsible for nanoparticle stabilization, while the difference in electrochemical potentials is the main reason for the ionic silver and phytocomponent interaction. In addition, the highly negative Zeta potential value demonstrated that AgNPs are capped by the bioactive compounds with a negative charge. These biomolecules with negative charges could adsorb on the surface of AgNPs and increase their stability via electrostatic repulsion and steric effect. Zeta potential is an indicator of the magnitude of electrostatic interactions between dispersed particles and can indirectly evaluate the stability of nanoparticles in colloidal suspensions by reflecting their ability to electrostatically repel each other. However, the change in UV-Vis spectra is indicative of the aggregation and formation of larger AgNPs in the presence of NaCl. The Zeta potential further confirmed the results of UV-Vis spectra. The added Na^+^ might bind with these biomolecules, which makes the AgNPs unstable [42]. The Zeta potential values of AgNPs synthesized with *E. camaldulensis* and *T. arjuna* were −26 mV ± 4.61 mV and −20 mV ± 5.09 mV, respectively. This negative Zeta potential value may be due to the potential capping of the bioorganic components present in the plant’s extracts. The high negative values reveal the electrostatic repulsion between the particles and facilitate the achievement of stable AgNPs without any agglomeration [42].

The pH is one of the key parameters promoting successful biosynthesis, often requiring high pH values. For example, *Annona reticulata* extract may contain molecules with a higher degree of protonation, so it is not able to reduce silver nitrate to AgNPs at pH values around 6.5–7.3. But when NaOH is incorporated with molecules in the extract, it ionizes the molecules present in the extract, altering the particle charges between the molecules and AgNO_3_ [54]. By virtue of the high concentration of hydroxyl groups on the nanoparticle surface at an alkaline pH, the size of the nanoparticles decreases due to the prevailing repulsive forces in colloidal solutions, leading to a decrease in aggregation [55]. At an acidic pH, large nanoparticles form due to the aggregation of AgNPs. On the other hand, at an alkaline pH, many functional groups are able to bind silver ions, making it possible to obtain smaller AgNPs. Thus, for the synthesis of nanoparticles using *Salvia verticillata* and *Filipendula ulmaria*, the pH influence on the AgNPs had the same effect: at pH 11, the synthesis was the best, while at pH 3, nanoparticles were practically not formed [56]. The pH can be involved in ionization by electron transfer under alkaline conditions when other compounds are present in an aqueous solution along with silver salt that contributes to AgNP formation [57]. Even higher pH (13) is required for AgNP biosynthesis by *Erythrina abyssinica* [55].

The temperature is equally important. It is known that high temperatures promote nucleation, while low temperatures promote growth. For example, the optimal temperature for phytosynthesis by *C. fragrans* and *E. abyssinica* was 80 °C [55]. In addition, AgNPs can maintain stability at high temperatures up to 100 °C [58]. There is also information about synthesis at lower temperatures, such as 25 °C for *Prosopis juliflora*-mediated nanoparticles [59] and 49.8 °C for *Lepechinia meyenii* (salvia) [60]. The optimal temperature for the synthesis by *Crocus haussknechtii* Bois was 75 °C [61]. At higher temperatures, the AgNP size increases due to biomolecule destruction or exceeding the boiling point during extraction because some biomolecules can evaporate at high temperatures [61]. In addition, 70 °C was also the optimal temperature for synthesis using *Teucrium stocksianum* [62]. A temperature of 60 °C was necessary for the effective AgNP synthesis by *A. indica* (Neem plant) [63]. At higher temperatures (50–60 °C), the formation of smaller nanoparticles in *Loranthus pulverulentus*-mediated biosynthesis occurred more rapidly [64]. Apparently, the differences in optimum temperature during AgNP synthesis by various plant extracts are primarily related to the compounds involved in their formation and the temperature required for their efficient operation.
antibiotics-14-00005-t002_Table 2Table 2Plant-mediated AgNPs.PlantSourceShapeSize, nmRef.*Euporbia serpens*Leavesspherical30–80[38]*C. fragrans*Leavesspherical48[39]*S. hispanica*Seeds spherical7[40]*Swertia paniculata*Aerial partsspherical31–44[41]*T. arjuna*Barkspherical2–100[42]*Atropa belladonna*Mother tincturespherical15–20[43]*E**. camaldulensis*Leaves spherical100[42]*Camellia sinesis*Leavesspherical10–16[44]*Litchi chinensis*Leavesspherical40–50[45]*Ferula persica*Aerial partsspherical15[46]*Buddleja globosa*Leavesspherical16[47]*Linum usitatissimum*Leavesspherical~47[48]*Phyllanthus amarus*Leavesflower-like30–42[49]*Ricinus communis*Roots,Leaves spherical2937[50]*Momordica charantia*Leaves spherical16[51]*Protium serratum*Leaves spherical~74[52]*Citrus maxima*Peel spherical4–11[53]*A. reticulata*Leaves cubic6.5–8.13[54]*E. abyssinica*Leavesspherical8.4–10[55]*Wedelia urticifolia*Flower spherical30[57]*Moringa oleifera*Leaves spherical10–25[58]*Prosopis juliflora*Bark spherical~55[59]*Lepechinia meyenii*Leaves spherical40–60[60]*Crocus haussknechtii Bois*Bulbs spherical10–15[61]*Teucrium stocksianum*Aerial partsspherical61[62]*Azadirachta indica *(Neem plant)Leaves spherical22–30[63]*Loranthus pulverulentus*Leaves spherical8–15[64]

### 2.5. Capping Agents

Capping agents are an essential component in the synthesis and stabilization of AgNPs. They act in several ways. First, they prevent the aggregation of nanoparticles. Second, biological molecules are primarily responsible for various medicinal properties. This increased activity of AgNPs is associated with a biological corona formed by metabolites from the “bio-factory”. This corona contains various compounds, such as proteins, enzymes, and plant metabolites. These “green” molecules increase the probability of attachment and exposure to microbial cells. The corona layer can promote the assimilation of nanoparticles, help interact with the pathogen membrane, enhance antimicrobial action and reduce toxicity to animal cells, and enlarge their long-term stability and biocompatibility [3]. Having their own antibacterial potential, bio-capping agents promote the synergistic effect of the nanoparticles and their covering layer. These agents surround Ag^0^ atoms, possibly by electrostatic attraction, Van der Waals forces, or covalent bond formation, giving AgNPs their characteristic shape and size.

The proteins in biologically synthesized AgNP solutions contribute to nanoparticle stabilization. Protein components in the medium can bind to AgNPs through free amino or cysteine groups, acting as capping agents that favor nanoparticle stability in solution and prevent their agglomeration [11]. Proteins as capping agents are typical of bacterial, fungal, algal, and plant synthesis. Proteins with the ability to interact with specific biomolecules may be responsible for biocompatibility and enhanced antimicrobial activity were characterized in the biosynthesis by *P. aeruginosa*. Cold shock protein, uncharacterized protein PA1579, Phospholipid-binding protein MlaC and PhoP/Q and low Mg^2+^ inducible outer membrane protein H1 were shown [65]. The 85 kDa protein operates as a coating material and gives stability to the fungal AgNPs. Two proteins (36 and 40 kDa) were detected in the extract and AgNPs synthesized by *Trichoderma harzianum*, which were identified as β-1,3-glucanase and chitinase enzymes, respectively [23]. In the synthesis of AgNPs by *Aspergillus flavus*, a 32 kDa protein, participates in the Ag^+^ reduction, and a 35 kDa acts as a capping agent for nanoparticles [3]. Most AgNP-associated proteins were involved in the redox mechanism in algal cells, the activity of ATPase, sedoheptulose-1,7-bisphosphatase, carbonic anhydrase, Superoxide dismutase (SOD), oxygen release enhancer protein, ribulose-bisphosphate carboxylase, and nuclear histone (H4). *Aloe vera* extract proteins act as capping agents to prevent agglomeration and provide stability in the medium [66]. Peptide bonds in proteins and enzymes present in *T. arjuna* leaf extract are broken down to form smaller peptides. Carboxylate groups in these peptides have a high affinity for the action as surfactants and form a protein corona layer on AgNPs, stabilizing them [42].

Typically, exopolysaccharides contain negatively charged functional groups, such as amide, carboxyl, and hydroxyl hydroxyls, which are able to interact with metal cations. Exopolysaccharides including D-glucose, L-fucose, D-mannose, D-galactose, N-acetyl-D-glucosamine, and N-acetyl D-glucosamine, can play as capping agents for *P. putida* and *E. coli* [67]. Polysaccharides also have the possibility of being capping agents in apple pomace extract and *Bersama engleriana* fruit extracts [68]. Antibacterial activity is a notable feature of some polysaccharides, as demonstrated by the synthesis of *Acacia rigidula* and *Rhodotorula mucilaginosa*. The effect of these polysaccharides on the antibacterial and anti-biofilm activity against both Gram-positive and Gram-negative bacteria has been demonstrated [69].

Flavonoids are known for their antimicrobial effects, realized in various ways. They damage the cytoplasmic cell membrane, inhibit enzymes vital to the microbial cell, and decrease the synthesis of fatty acids and membrane lipids. Therefore, their participation as capping agents seems to be especially relevant for enhancing nanoparticle antibacterial effects. Flavonoid amide groups have a strong affinity for metal ions and can encapsulate nanoparticles, forming a protective shell that prevents further aggregation during their synthesis into *Annona reticulata* [54]. AgNPs were found to be capped with flavonoids for *Dypsis lutescens* [70] and *Carduus crispus* [71]. The flavone compounds of gallic acid, cianidanol, and epicatechin gallate from grape seed extract have adequate antibacterial activity against bacterial strains such as *E. coli*, *P. aeruginosa*, *S. aureus*, and *B. subtilis*, which can play the capping agent role [72]. Flavonoids can bind to Ag^0^ in *Calotropis gigantea* by forming a complex because these compounds contain a chelating part, catechol [73]. Nanoparticles synthesized by green tea can be capped by epicatechin [74]. One of the most prevalent phenolic compounds detected in *Rumex* sp. is epigallocatechin gallate, which can also act as a capping agent [75].

The layer of biomolecules covering nanoparticles can be represented by tannins, which are also known as antibacterial agents. It contains a hydrophilic functional group for hydrogen bonding. The hydrophobic moieties and hydrophilic shells of tannic acid stabilized on AgNPs play an important role in its interaction with the hydrocarbon chain of lipids, as well as surface proteins on bacterial cells. Tannins are capable of inhibiting the growth of bacterial and fungal cells, disrupting the permeability of microbial membranes and cell walls. They also inhibit cell protein synthesis through a protein-binding mechanism. Algal tannin can serve as a capping agent [76]. A similar effect was found in tannins extracted from *Hyssopus* and *Calendula*, as well as *E. suberosa* [77].

Some alkaloids may have direct antibacterial activity, and some may enhance the antibiotic effect. These properties make them potentially useful substances capable of playing the capping role. AgNPs synthesized by *Murraya koenigii* can contain carbazole alkaloids that have potent antibacterial activity [78]. Belladonna alkaloids surround the nanoparticles and act as the capping agent for the AgNPs [43].

Other biomolecules of plant origin, such as saponins, and terpenoids are able to form part of the coating layer for AgNPs. Saponin antimicrobial activity and its synergistic effect with antibiotics against various pathogens are known. The capping role of saponin was noted for AgNPs synthesized by flaxseed extract of *Vernonia amygdalina* [79]. Energy dispersive X-ray spectroscopy chemical analysis showed that chamomile terpenoids act as capping agents and are adsorbed on the surface of chamomile extract-mediated AgNPs, preventing their agglomeration [80]. The terpenoids as covering biomolecules were shown for *Stachys parviflora* AgNPs [81]. The oxidized form of polyphenolic terpenoids can bind the AgNPs via –C=O thus causing both reduction and stabilization. Concentrated terpenoids could be adsorbed on the surface of metal nanoparticles by interaction through π-electrons or carbonyl groups in the absence of other strong ligating agents. Different biological capping agents from various organisms are presented in Figure 3.

Thus, the biological compound variety involved in the AgNP “corona” formation may be one of the most important aspects that form the antibacterial properties of AgNPs. The production of nanoparticles with desired properties therefore proposes the characterization of these compounds to play the role of capping agents for their future medical use.

## 3. Mechanism of Action

Although the mechanism of AgNP synthesis using “bio-factories” has not been fully studied, there are currently many data on this subject. Most researchers note several areas through the antibacterial effect of AgNPs is realized, such as (1) cell membrane destruction and leakage of its cellular contents; (2) binding to functional groups of proteins, causing protein denaturation and cell death; (3) penetration inside the cell and damage of intracellular structures (vacuoles, ribosomes); (4) inactivation of the respiratory chain; (5) blocking DNA replication and/or turning it into a condensed form; (6) alteration in the transcription processes; (7) inhibition of protein biosynthesis (translation); (8) denaturation of enzymes transporting nutrients through the bacterial cell membrane; (9) generation of reactive oxygen species (ROS); (10) modulation of signal transduction pathways (Figure 4). According to these ways, AgNPs are able to destroy a wide range of microorganisms, including both Gram-negative and Gram-positive, making them a wonderful remedy even against antibiotic-resistant strains of bacteria.

Adhesion to the surface of the bacterial cell wall and the membrane is one of the most significant mechanisms involved in the AgNP antibacterial activity. The AgNP’s ability to attach to the bacterial cell wall is due to electrostatic interactions between positively charged silver ions and negatively charged cell membrane surfaces due to carboxyl, phosphate and amino groups, which make it possible to subsequently penetrate it, thereby causing structural changes in the cell membrane and, as a result, its permeability [3]. Differences in the effects of AgNPs on Gram-positive and Gram-negative bacteria were discovered at the dawn of research. The main reason for these differences is the structural distinction in the cell walls of these two types of bacteria. The cell wall of Gram-negative organisms is represented by a thin layer of peptidoglycan, consisting of linear polysaccharide chains crosslinked with short peptides, thus forming thinner structures that facilitate the easy penetration of nanoparticles, while the cell wall of Gram-positive microorganisms has a thick layer of peptidoglycan. The adhesion and accumulation of AgNPs on the cell surface were particularly characteristic of Gram-negative bacteria. AgNPs can enter bacterial cells through water-filled channels called porines in the outer membrane of Gram-negative microorganisms. Porines are mainly involved in the passive transfer of hydrophilic molecules of various sizes and charges across the membrane. Thus, after an hour-long incubation of *E. coli* with AgNPs produced by chicory extract, an increase in surface roughness and, in some cases, membrane destruction and leakage of intracellular contents were observed, while structural changes in *S. aureus* were less notable [82]. Besides that, capping agents—aromatic/hydrophobic components of the main phenolic compounds in chicory extract adsorbed on AgNPs—apparently promote interaction with the lipid hydrocarbon chain, leading to close contact between this nanomaterial and the lipid surface [82]. It is possible that the lipopolysaccharide presence facilitates the structural integrity of the Gram-negative bacterial cell wall, making such bacteria more sensitive to AgNPs because the negative charge of lipopolysaccharides promotes AgNP adhesion. The lipid membranes of Gram-negative bacteria may also be one of the factors stimulating better nanoparticle penetration The interaction of AgNPs with model lipid membranes did not in itself cause a disruption of the lipid structure. However, AgNPs accumulating on the cell wall can act as a local silver reservoir, or as a Trojan horse, releasing a sufficiently high concentration of toxic Ag^+^ ions into or near a bacterial cell. AgNP accumulation on the cell membrane creates ruptures, and disrupts the layer integrity, resulting in an increase in permeability and, ultimately, bacterial cell death [82]. Similar data were obtained for *P. aeruginosa*, where cells have entire open structures and porous membranes and are surrounded by nanoparticles on their surface [66]. The authors also suggest that AgNPs produced by *P. putida* are able to easily attach to membranes and cause cell lysis due to related capping agents [66]. In addition, nanoparticles seem to be capable of forming “pits” on the cell surface, and NPs accumulation occurs in the cell. Under the influence of AgNPs created by *Litsea cubeba*, *E. coli* cells were deformed and destroyed, and some began to atrophy [83]. It was found that, after contact with *Viridibacillus* sp.-AgNPs, the *E. coli* and *P. aeruginosa* cell membranes are completely destroyed. The AgNP attachment to the negatively charged surface of the cell wall and membrane induces cytoplasmic compression and membrane detachment, ultimately leading to cell wall rupture. Moreover, AgNP interaction with sulfur-containing proteins located in the cell wall can also affect membrane permeability and cause cell leakage [83]. A polyphenol-rich plant extract of *A. indica* formulates an intact, stable polyphenol–nanoparticle conjugate that can adsorb oxygen molecules and release silver ions to adhere to the *E. coli* membrane, contributing to cell destruction [84]. The *E. coli* cells treated with AgNPs from *Sargassum swartzii* showed unevenness and a loss of rigidity [85]. On ultrathin slices of *P. aeruginosa* cells treated with *F. solani*-AgNPs, recessions in the cell wall with AgNP penetration, fragmentation, complete disappearance of cellular contents, disorganization, and leakage of internal components were found [26]. In the case of *S. aureus*, Gram-positive bacteria, AgNPs destroy the β-1,4-glycoside bond between N-acetylteichoic acid and N-acetylglucosamine in the bacterial cell wall, resulting in damage to the multilayered reticular structure of peptidoglycan and ultimately adducing the cell wall cleavage and the N-acetylglucosamine release [86]. The *Streptococcus mutans* cells treated with AgNPs had a smaller size and an irregular shape compared to the control and also died after 24 h. However, *P. aeruginosa* and *E. coli* died after only 4 h of treatment [87]. The *S. aureus* control cells remained transparent and in cocci form without any changes in cell morphology, while cells processed with AgNPs from the *Rheum palmatum* root extract showed membrane swelling and deformation [88].

It should be noted that AgNP size has a considerable value, because smaller NPs have a larger surface area available for interaction and penetration into internal cells. Singh and Mijakovic propose that larger AgNPs may cause damage to the membrane, on the contrary, smaller nanoparticles (less than 10 nm) can penetrate cells after adhesion and injure intracellular structures, affecting vital cellular functions [84]. Additionally, spherical nanoparticles exhibit greater antimicrobial potential compared to nanoparticles of other shapes [83].

The surface potential of a bacterial cell plays an important role in maintaining cell growth and other metabolic processes. It also provides valuable information about membrane integrity and other surface characteristics. A change in the membrane potential leads to a change in its permeability, cellular component leakage and subsequent cell death. The interaction between AgNPs and bacterial cells is also shown by a significant decrease in the Zeta potential of the cell surface for both Gram-positive and Gram-negative bacteria. Treatment with biogenic AgNPs from *E. coli* and *B. subtills* led to a decrease in cell surface potential indicating a significant level of membrane modification by AgNPs. Bacterial cells became hollow, deformed, and compressed after 4 h AgNP exposure, demonstrating serious damage to their membranes, followed by the release of cellular components into the medium. *E. coli* cells treated with AgNPs created using *Camellia sinensis* leaves showed signs of damage, including cracks and wrinkles on the outer surface and completely deformed cell membranes. Destruction of cell membranes can be triggered by AgNP accumulation in the bacterial membrane, causing a change in the membrane potential and stimulating pit formation in the membrane and additional cell lysis, ultimately leading to cell death [44].

Interacting with a cell, nanoparticles trigger morphological changes in membrane structure, provoking damage to membrane permeability and respiratory function by virtue of membrane depolarization and, eventually, to a breach of cell integrity and death. Ultrastructural changes, such as bacterial deformation, cell shrinkage, and cell membrane thinning, were observed in *K. pneumoniae* influenced by AgNPs fabricated using *Nostoc* sp. [89]. The complete separation of the plasma membrane from the bacterial cell wall and the appearance of atrophied cells were accompanied by membrane folds and ruptures, nucleoplasm agglutination, cytoplasm binding, and the release of cytoplasmic contents into the environment. *P. aeruginosa* affected by AgNPs from *A. flavus*, showed significant damage, indicating nanoparticle accumulation on the cell wall and inner and outer bacterium poles. The cell wall had wavy and discontinuous outlines, and the plasma membrane had ruptures. Moreover, accumulations near the plasma membrane, probably chromatin, were observed, and the cytoplasm electron density decreased, testifying to the loss of cellular material [90]. Mechanical damage to cell membranes, changing their surface stiffness, roughness and adhesion ability, was observed in *S. aureus* under the AgNP impact. It is supposed that impaired membrane function could provide changes in structural and mechanical membrane properties, resulting in bacteria’s non-viability. *S. aureus* and *B. subtilis* treated with green AgNPs showed signs of pore formation on the outer membrane, confirming the membrane damage. The mechanism of silver action is related to its interaction with thiol group compounds in bacterial respiratory enzymes. AgNPs attach to the cell wall and membrane, suppressing the respiratory process [91]. The high affinity of silver particles for sulfur and phosphorus, which are found abundantly throughout the bacterial cell membrane, may be a mechanism for antibacterial activity. Nanoparticles react with proteins containing sulfur either inside or outside the cell membrane, affecting cell survival [91].

The assimilation of nanoparticles by cells can be realized in two stages. First, the binding stage takes place on the cell membrane, and then the nanoparticle absorption by endocytosis takes place. In the endocytosis process, cells easily absorb nanomaterials through the invagination of a small surface area to form a new intracellular bubble around the substance, which is transported into the cells. *E. coli* and *S. aureus* were found to have successfully absorbed AgNPs. AgNPs, by disrupting the outer membrane permeability, cause leakage of cellular materials, penetrate into the inner membrane, and produce ROS, thereby suppressing cell growth. At the same time, AgNPs can affect some cellular components, ultimately leading to cell death [91]. Cell membrane damage of *Salmonella enterica* owing to AgNPs from the *P. aeruginosa* biofilm layer promotes the rupture of some sites, and intracellular components leave the cell [92]. The increase in compound content, such as amino acids, polyamines, and organic acids, in cells treated with AgNPs from *Tinospora cordifolia*, compared to untreated cells proves the effectiveness of AgNPs, which destroys bacterial membranes [93]. Interestingly, *E. coli* also secreted various L-amino acids that act as osmolytes and, thus, could allow bacteria to survive under AgNP influence [93]. It is very likely that capping agents, such as tannins and flavonoids, have the ability to form complexes with membrane proteins, initiating the destabilization of bacterial membranes. Ultimately, nanoparticles can cause irreparable damage to cell membranes and accumulate in the cytoplasm, invoking cell death.

Another possible mechanism of antimicrobial activity is due to free radical formation when AgNPs interact with bacteria, resulting in the release of silver ions inside the cell, causing the cell contents to leak out and ultimately initiating protein denaturation [53]. ROS are by-products of normal oxygen metabolism, and their level is controlled by the antioxidant protection system, such as glutathione/glutathione disulfide (GSH/GSSG). ROS are formed by interactions with enzymes and/or biomolecules, causing cell damage. Excessive production of ROS, such as superoxide, hydrogen peroxide, and hydroxyl radicals, often promotes cellular oxidative stress in microorganisms. Therefore, the ability of AgNPs to generate ROS and free radicals and thus enhance oxidative stress in cells is often mediated by their antibacterial activity [94]. The production of ROS could be caused by the impeded electronic transport along the respiratory chain in the damaged plasma membrane. Respiratory enzymes are inhibited by the interaction of metal ions with the -SH groups and also by the auto-oxidation of NADH dehydrogenase II in the bacterial respiratory chain. In the extracellular space, AgNPs can induce the formation of oxygen free radicals (O^−^) and superoxide free radicals (O^2−^); on the cell membrane, AgNPs probably induce the electron penetration of membrane-bound respiratory chain enzymes into the cytoplasm and react with cytoplasmic oxygen with the formation of superoxide radicals (O^2−^). As a result, free radicals, or ROS, can attack membrane lipids, and disrupt the function of membranes, and can also evoke conformational changes in membrane proteins and DNA structure, causing cell death [95]. Studies conducted with AgNPs from *Ocimum gratissimum* leaf extract showed that ROS level increase may be one of the factors responsible for bacterial cell death. Treatments with AgNPs have been shown to increase ROS generation in *S. aureus* and *E. coli*, damaging the bacterial cell membrane and protein structure, as well as the intracellular system. ROS generation analysis under the influence of *Capsicum annuum*-mediated AgNPs on Gram-positive and Gram-negative bacteria, *P. aeruginosa* and *S. aureus*, revealed that the latter produced more ROS, according to differences in the architecture and composition of bacterial cells [95]. It was shown that “green” AgNPs facilitated ROS generation *in S. mutans* and *Actinomyces viscosus*, which in turn can lead to disruption of biomolecules and organelle structure, oxidative carbonylation of proteins, lipid peroxidation, DNA/RNA rupture, and membrane structure damage [96]. It was determined that AgNPs induce excessive ROS generation in multidrug-resistant *P. aeruginosa*, depending on time and concentration. Apparently, ROS production occurs during the early stages of interaction between AgNPs and bacteria.

AgNPs can reduce the expression of antioxidant enzymes like glutathione (GSH), superoxide dismutase, and catalase, which can accelerate ROS accumulation. Disrupting purification mechanisms, AgNPs bind directly to thiol groups in the corresponding enzymes and to GSH in its oxidized form, glutathione disulfide (GSSG), thereby increasing ROS and free radical concentrations.

The reaction to oxidative stress is a different degree of macromolecular substance oxidation, such as DNA and proteins. Comet assay, a sensitive approach to quantifying DNA damage at the level of an individual cell, was used to assess bacterial DNA damage induced by ROS. The results showed that ROS produced by bacteria could cause DNA damage to *Vibrio natriegens*, and the higher the concentration of ROS, the more severe the DNA damage [97]. Apart from that, smaller-sized nanoparticles produced more ROS in bacteria. AgNPs from *Citrus latifolia* and *A. flavus* stimulated ROS-induced peroxidation and hydroxide radical damage to DNA, causing double-stranded breaks and activating RecA (RecA is a protein essential for the repair and maintenance of DNA in bacteria). Reishi mushroom-mediated green synthesized AgNPs have DNA cleavage activity and inhibit microbial growth by this mechanism [98]. Similar data were obtained for AgNPs from *Datura stramonium* leaf extract: the formation of DNA strand breaks probably correlates with silver ion release, which strongly depends on the nanoparticle size, where smaller particles cause more damage than larger ones. In addition, the authors suggest that alkaloids play the capping agent’s role and may be of great importance for antibacterial activity [98]. Sulfur and phosphorus are important components of DNA, their interaction with AgNPs can inhibit DNA replication and cell reproduction, finally encouraging the microorganism’s death. Moreover, Ag^+^ can form complexes with nucleic acids, breaking H-bonds between antiparallel base pairs. A change in the DNA molecule’s state from a relaxed to a condensed form can also be caused by AgNPs. Reduced ability to replicate is another consequence of the incorporation of AgNPs into DNA. The transcription process in microorganisms can be inhibited by embedding AgNPs into the DNA helix. AgNPs can interact with DNA bases, forming cross-links and replacing hydrogen bonds associated with nitrogen atoms in purines and pyrimidines. Nanoparticles, mediated by *Bersama engleriana*, induce DNA release into the extracellular environment, which correlates with binding to nuclear DNA in MRSA [68]. AgNP binding to DNA can provide DNA denaturation and disruption of cellular division, and the formation and accumulation of damaged DNA can have a significant effect on protein synthesis. It was shown that AgNPs synthesized by *Helianthemum Lippii* have a strong tendency to bind to DNA through electrostatic interactions, as evidenced by the binding energy values [99]. The replication efficiency of small (600 bp) and large (1500 bp) DNA fragments in the presence of *Thalictrum foliolosum* root extract-mediated AgNPs decreased in a dose-dependent manner, and non-specific adsorption of DNA polymerase by nanoparticles was observed [100]. Silver ions can also interact with phosphorus residues in DNA, causing inactivation of DNA replication, where DNA damage was shown using a reporter strain and *A. flavus*-AgNPs as an example [89]. Interestingly, Van der Waals and hydrogen bonding interactions were responsible for the interaction between AgNPs from *Epipremnum aureum* and calf thymus DNA (CT-DNA) [101].

There is also evidence of the AgNP interaction with cytoplasmic structures, including ribosomes, affecting translation and transcription. Silver ions can bind to the 30S ribosome subunit, deactivating the ribosomal complex and stopping protein synthesis. The synthesis of immature precursor proteins that are crucial for the formation of the cell membrane is disrupted by the AgNP action on ribosomes, transcription, and translation, leading ultimately to cell death [3].

Both Ag and Ag^+^ can accept lone pair electrons to form coordination bonds or to have strong electrostatic attractions. The interaction between silver ions and thiol groups in enzymes, as well as other groups such as -NH, -OH, -SH, -COOH, and -SS, occurs through weak hydrogen bonds. This is an interesting finding, because, according to proteomic analysis, when exposed to chemically synthesized nanoparticles on *P. aeruginosa*, membrane proteins were identified, whose expression appears to be regulated by AgNPs. These proteins are primarily involved in ion binding, transport, flagellar assembly, pore formation, antibiotic resistance, and membrane stabilization. Three major regulated proteins (atpE, PA2536, and PA4504) play a role in ATP synthesis, phospholipid synthesis, and transmembrane transport. Some outer membrane porins (OprH, OprD and OprC) associated with the transfer of cationic amino acids, peptides, antibiotics, and ions were significantly affected by AgNPs. Metal transporters in *P. aeruginosa* were suppressed by AgNP impact, potentially promoting the transfer of AgNP and/or Ag^+^ through transmembrane pores. Flagellins playing an important role in motor activity, adhesion, biofilm formation, and phage infection, were also influenced by nanoparticles. This suggests that AgNPs interact with the cell membrane and proteins, leading to the release of silver ions [102]. Proteomics analysis has shown that in *P. aeruginosa* bacteria treated with AgNPs, the levels of SOD, CAT (catalase), and POD (peroxidase), as well as alkylhydroperoxide reductase and hydroperoxide resistance proteins, are significantly increased. Additionally, there is an increase in the activity of oxidases that regulate low oxygen content, such as cbb3-type cytochrome c oxidase. The analysis of protein interactions showed that the levels of most oxidative and antioxidant proteins were increased under nanoparticle influence. However, the protein number involved in DNA and RNA damage, as well as ribosomal proteins, decreased. Therefore, AgNPs can stimulate oxidative stress in *P. aeruginosa*, and reduce the local oxygen pressure, which inversely upregulates the corresponding reductases and hypoxia regulatory oxidases, and downregulates the constitutive and hyperoxic regulatory oxidases. Ultimately, AgNPs can influence the redox processes in DNA replication, RNA transcription, biosynthesis, and metabolism of ribosomes, purines, pyrimidines, and fatty acids in bacteria.

It is very likely that bionanoparticles can act in a similar way. When producing nanoparticles using lignin, upregulation was detected in the genes *tonB* (membrane protein, transport of various compounds), and *luxR*, while, *mfs* (membrane transport, efflux of various compounds) and cap (capsule biosynthesis) were found to be downregulated, and PelB (biofilm formation through a transmembrane protein) experienced only a slight increase in expression. Thus, genes encoding membrane proteins with an efflux function were upregulated. However, all other genes were membrane proteins that did not efflux metals and were downregulated [103]. Analysis of the effect of AgNPs from *Lactobacillus acidophilus* showed that induced genes may be transcription factors, such as zinc finger protein (it has multiple functions as virulence, symbiosis, and/or cell cycle transcription), and genes that are completely suppressed are mainly DNA protection or resistance genes, such as the *mecA* gene [104].

Interesting data were obtained during the treatment of Gram-positive and Gram-negative bacteria with AgNPs produced by *B. licheniformis*. After incubation, protein leakage from AgNP-treated cells increased significantly. Moreover, more proteins leaked through the membranes of *S. aureus* than through those of *E. coli*, suggesting that the antibacterial sensitivity of *S. aureus* is higher than that of *E. coli* [105]. At the same time, increased time and concentration led to greater protein release in *P. aeruginosa* compared to *B. subtilis* [105]. Additionally, AgNPs were found to enhance the membrane leakage of reducing sugars from bacterial cells and inhibit the activity of respiratory chain dehydrogenase. Protein leakage was observed after AgNP influence in *S. mutans* and *A. viscosus* [96]. The leakage of reducing sugars and proteins from the cytoplasm was found to be more significant for *E. coli* than for *B. cereus* under the influence of *Tectona grandis*-AgNPs [106].

It is assumed that AgNPs can penetrate the outer membrane barrier, peptidoglycan, and, periplasm, and destroy dehydrogenases in the respiratory chain, thus suppressing cell respiration. AgNPs prevent bacteria from utilizing nutrients present in the nutrient medium by inhibiting the production of enzymes involved in nutrient metabolism. Such an obstacle can lead to a decrease in bacterial survival, as the pH of the bacterial cytoplasm is controlled by the protons passing through the respiratory chain and the influx of potassium ions (K^+^), and also because the ion exchange systems of bacteria are associated with ATP energy synthesis [107]. AgNPs synthesized by aqueous extracts of papaya seeds, roost, and bark inhibited H^+^-ATPase proton pumps in *E. coli* [107].

Inhibition of ATP synthesis may be one of the important antimicrobial mechanisms of AgNPs. A significant decrease in ATP levels in bacterial cells was demonstrated after treatment with different concentrations of *Pilimelia columellifera*-AgNPs [108]. Cellular stress caused by AgNPs affects ATP synthesis, which, in turn, affects bacterial growth and reproduction negatively. It was also found that AgNPs affected FOF1-ATPase activity and H^+^-coupled transport in the bacterial cell membrane. In other words, this membrane ATPase is a potential target for AgNPs, and its suppression can impact the metabolic processes in bacteria.

Numerous studies about AgNP synthesis, including those conducted using well-known medicinal plants, allow the conclusion that capping agents, according to their own antimicrobial activity, are involved in the antibacterial mechanism of AgNPs. Tannins, for example, acting as such agents in nanoparticles from *E. camaldulensis*, contribute to enhanced antibacterial properties because these compounds are known to inhibit cellular protein synthesis through a protein binding mechanism. The delivery of polyphenolic compounds from tea extracts to microbial cells on the surface of nanoparticles can enhance the antimicrobial effect. This phenomenon was described as a “poisoned arrow” mechanism [109].

The antibacterial activity of biological AgNPs was studied with respect to a wide range of microorganisms. Some examples are shown in Figure 5.

When discussing alternatives to AgNPs as a replacement for well-known and commonly used antibiotics, it is important to note that pathogens can develop resistance via the following mechanisms: (a) modifying target proteins; (b) enzymatically degrading or inactivating the drug; (c) reducing the permeability of cell membranes, which blocks drug entry; and (d) enhancing the excretion of the drug [82]. That is why the potential applications of AgNPs are very diverse, and the mechanisms of resistance development in pathogens are intricate.

### 3.1. Antibiofilm Activity

An accumulation of bacteria with a strong ability to survive in harsh environmental conditions and resist host immunity and various therapeutic drugs forms biofilms. These microbial communities, relatively insulated from the outside world, consist of one or more types of bacteria enclosed by an extracellular polymeric substance, primarily made up of polysaccharides, nucleic acids, and proteins. Thanks to biofilms, many pathogenic microorganisms are highly resistant to drugs. Up to 65% of chronic human infectious diseases may be linked to biofilms. Therefore, developing new drugs that can inhibit biofilm formation is crucial in the fight against the increasing resistance of human pathogens to existing medications.

Six main components can be identified in biofilms: polysaccharides, proteins, nucleic acids, lipids, water, and ions (such as cations like Ca^2+^ or Mg^2+^, and anions like Cl^−^ or PO_4_^2−^) [110]. The component composition varies depending on the type and species of bacteria present in the biofilm. These components help the biofilm adhere to biological and non-biological surfaces, contribute to its maturation, and maintain its architecture and viscoelasticity along with water and ions. Polymers, along with ions and water, contribute to the maturation and maintenance of the architecture and viscoelasticity of biofilms. Polysaccharides and biofilm proteins restrict the movement of newly dividing bacterial cells inside the biofilm, keeping them in close contact with each other [110]. The enzymes present in large quantities catalyze lytic and redox reactions necessary for nutrient availability. They also help remove the remnants of dead cells and destroy toxic molecules in the biofilm microenvironment. Due to lipids, the biofilms gain hydrophobic properties, which are essential for adhesion, the absorption of hydrophobic nutrients, and the maintenance of surface tension at the interface [110].

Biofilms are involved in maintaining and activating dormant bacteria, leading to increased pathogenicity and the progression of infection or disease [110]. They are also resistant to many antibiotics, due to their ability to inhibit antibiotic spread through molecular sieves and interconnected extracellular polymeric substances (EPS), enzymatic degradation, increased hydrophobicity (for example, secretion of BslA protein in *Bacillus* sp.) and genetic modification.

Combined methods using various antibiotics help in the destruction of biofilms. For example, clarithromycin along with vancomycin was found to destroy the biofilm-forming bacterial cells. This combination is active against Gram-negative bacteria and has specifically proven effective against *P. aeruginosa* and *Staphylococcus* sp. This combination targets the major component of the EPS matrix, i.e., the alginate, which is thick, solid and difficult to destroy, preventing the entry of antibiotics. Another combination of a macrolide and a carbapenem, i.e., roxithromycin and imipenem helps white blood cells penetrate inside the matrix and destabilize the biofilm, eventually eradicating it. The combination of clarithromycin and levofloxacin with an efficacy rate of 99% was effective against *P. aeruginosa* compared to individual antibiotic therapy [110]. The combination of N-Acetylcysteine NAC (4890 μg/mL) and ciprofloxacin (32 or 64 μg/mL) had a synergistic effect. This combination also showed antibiofilm activity against *P. aeruginosa* and other microbes. NAC was thought to inhibit EPS matrix production, which is one of the significant steps in destroying the rigidity of the biofilm. However, it is difficult for traditional antimicrobial agents to penetrate biofilm cells, as their diffusion is hindered by the structure of the biofilm. Exopolysaccharides act as a physical barrier, making it challenging for antibiotics to reach deeper layers of the biofilms. Additionally, the interaction between drugs and components of the matrix can slow down the rate of drug penetration. Drainage pumps, which are essential for normal bacteria function and biofilm formation, can displace antibiotic molecules from cells, leading to resistance. Furthermore, the proximity of cells in a biofilm can increase the likelihood of genetic exchange, including between different species, further increasing biofilm resistance to antibiotics [111].

The remarkable results obtained from the study of the antibacterial activity of AgNPs suggest that they also exhibit anti-biofilm activity. This phenomenon can be explained by the inhibition of biofilm formation and its components. The interaction between nanoparticles and biofilms takes place in several stages: AgNPs first interact with the surface of a biofilm, then penetrate it, and finally interact with biofilm and cellular components through electrostatic, hydrophobic, hydrogen bonding, and Van der Waals forces.

AgNPs from *Euphorbia hirta* are able to inhibit bacterial colonization on glass surfaces. In the absence of a matrix of exopolysaccharides, *K. pneumoniae* cells were found to be located in pairs or as several bacilli. Additionally, an increase in the roughness of the cell surface was observed, indicating damage caused by nanoparticles [112]. Thus, the treatment of medical devices with nanoparticles can prevent bacterial adhesion and biofilm formation. AgNPs, biogenically synthesized using *F. oxysporum* combined with oregano derivatives, have been shown to prevent biofilm formation and reduce the metabolic activity of preformed biofilms, resulting in a reduction in the biofilm viability by at least 95% and a decrease in total biofilm biomass of at least 90% in *E. coli* and *K. pneumoniae* [111]. Antibiofilm activity tests on fresh lettuce showed that nanoparticles from *Jacaranda mimosifolia* decreased the population of *S. aureus* and *P. aeruginosa* by 0.63 and 2.38 logarithms [113]. *Zataria multiflora*-derived AgNPs exhibited significant antibacterial activity, with a minimum inhibitory concentration (MIC) of 4 μg/mL against *S. aureus*. The dependence of the inhibitory effect of AgNPs on the biofilm formation by *C. violaceum*, *P. aeruginosa*, and *S. marcescens* was established. Treatment with AgNPs reduces bacterial adhesion and prevents the formation of a polymer matrix encapsulating the bacterial cells. The presence of AgNPs in the nutrient medium reduces bacterial colonization, scatters cells, and allows for observing a single layer of bacterial cells [78]. Treatment with respective 1/2× MICs of AgNPs synthesized using *Carum copticum* resulted in reduced adherence of cells to the glass surface; the treated cells formed small clumps of microcolonies with reduced biofilms [114]. Analysis of *E. coli* biofilms on urinary catheter segments under the influence of AgNPs derived from *Aloe vera* revealed that adhesion of *E. coli* to the catheter decreased linearly with an increase in the nanoparticle concentration (from 85 to 340 µg/mL) over time (up to seven days) [115].

As mentioned earlier, biofilms consist of microorganisms that adhere to a surface coated with mucus—an EPS. This matrix acts as a protective layer for microbes, protecting them from harmful environmental factors. Inhibition of EPS is an important virulence factor in the anti-biofilm formation action. It was found that nanoparticles affect *E. coli*, *K. pneumoniae*, *P. aeruginosa*, *Pr. mirabilis*, and *A. baumannii*, depending on their concentration. At low concentrations of AgNPs (12.5 µg/mL), microorganisms continue to grow, but nanoparticles inhibit the synthesis of the glycocalyx matrix. However, at higher concentrations, the nanoparticles suppress bacterial growth by more than 90%. When the glycocalyx synthesis stops, the bacteria can no longer form biofilms. High concentrations (100 µg/mL) completely blocked the formation of a biofilm and inhibited the growth of the organism. A study showed that AgNPs synthesized using *Prosopis juliflora* inhibit the secretion of exopolysaccharides, which are essential for biofilm formation [59]. *Abroma augusta*-AgNPs also demonstrated a dose-dependent effect on reducing biofilm formation at lower doses than those that inhibit methicillin-resistant *S. aureus* and vancomycin-resistant *Enterococci*. In addition, few polysaccharides were found, and growth was slowed [116]. The inhibition of EPS in *K. pneumoniae* by *Gracilaria corticate*-AgNPs was 80% at 90 μg/mL [117]. SEM analysis showed that the arrangement of hank-like structures containing exopolysaccharides was significantly damaged and quantitatively affected by exposure to AgNPs. Thus, nanoparticles interfered with the formation of biofilms and led to cell death [117].

For nanoparticles from *A. baumannii*, *E. coli*, and *P. aeruginosa*, 50% inhibition of *P. aeruginosa* biofilm was established, while *S. aureus* biofilms showed 50% inhibition at the same concentration only when treated with *S. aureus*-AgNPs. SEM showed that biosynthesized AgNPs were able to reduce the surface coating of *P. aeruginosa* and *S. aureus* biofilms and diminish the EPS, causing noticeable morphological changes in the biofilms [11]. Moreover, the cell surface swelling was observed in some cells, indicating the beginning of the apoptosis process in these cells. In *P. aeruginosa* cells treated with AgNPs at a concentration of 10 µg/mL, a decrease in colonization, loosely distributed cells, a disruption of biofilm structure, and a lower EPS were observed [11].

Bacterial growth, biofilm formation, and pathogenesis are all directly linked to bacterial motility. Swarming is crucial for surface adherence. Swarming motility and biofilm formation were inhibited by *Phyllanthus emblica*-AgNPs, indicating that the inhibitory effect of AgNPs is partly due to the disruption of flagella and interference with biofilm formation in *Acidovorax oryzae* [118]. AgNPs caused increased secretion of Hcp protein compared to the control in *A. oryzae*. Hcp is believed to be a component or effector protein, such as T6SS, which is a distinctive component of many Gram-negative bacteria; in turn, T6SS is associated with virulence and biofilm formation [119]. At the same time, it was found that T6SS activity was strongly induced due to membrane damage. Therefore, it is reasonable to assume that the increased secretion of Hcp effector proteins in *A. oryzae* can be partially explained by membrane disruption [119].

The inhibitory effect of AgNPs on an existing biofilm may be due to their ability to penetrate the EPS layer of the biofilm through water channels. Biofilms contain water-filled channels that are necessary for nutrient transport, and AgNPs can use these channels to directly interact with EPS and perform an antimicrobial function. It is possible that *C. copticum*-AgNPs diffuse through these water channels and produce antibacterial activity by disrupting the biofilm’s ability to transport nutrients. The thickness of the *K. pneumoniae*’s biofilm decreased under the influence of AgNPs, presumably due to their penetration into the biofilm via aqueous channels [117].

The *carO* and *bssS* genes encode small cytoplasmic proteins that play a significant role in regulating biofilm formation in *A. baumannii* and *K. pneumoniae*. In *P. aeruginosa*, the *pelA* gene is responsible for the synthesis and transport of polysaccharides, which are important components of the biofilm. AgNPs synthesized by *Streptomyces* isolates were found to significantly reduce the expression of these genes associated with biofilm formation [120]. AgNPs at 1/2× MIC concentration reduced the expression of *lasI* and *lasR* genes associated with biofilm production in *P. aeruginosa* [120]. The virulence factors for adhesion are type I fimbriae (FimH) and pillium structures for attachment to host cells. AgNPs downregulated the expression of *K. pneumoniae* virulence and biofilm-related genes *fimH*, *rmpA*, and *mrkA* [121].

The virulence factors of *A. baumannii* include the genes of adhesins like *kpsMII* (group 2 capsule synthesis) and *fimH*, *tratT* (serum resistance-associated), *fyuA* (yersiniabactin receptor) and *iutA* (aerobactin receptor). It has been shown that AgNPs are able to inhibit the expression of various virulence-related genes (*kpsMII* and *afa*/*draBC*) and biofilm-related genes (*bap*, *ompA*, and *csuA*/*B*). At a concentration of 25 μg/mL, AgNPs significantly reduce these genes expression [122].

Capping agents, such as antibiofilm activity stimulants, may also be important. Therefore, molecular docking has been used in [123]. This is a computational method that is widely used in biofilm control research to predict how small molecules interact and bind with target proteins involved in biofilm formation. The molecular docking study shows that the phytochemicals had a strong interaction with *Str. mutans* Antigen I/II carboxyterminus protein. Polyhydroxylated compounds were shown to have high interaction values with the investigated protein, which may be due to the potent H-bond interaction between ligands and proteins [123]. Various plant metabolites (in this case from *Valeriana jatamansi*) can act as capping agents for AgNPs, thereby increasing their antibiofilm potential.

### 3.2. Anti-Quorum Sensing Activity

The biofilm formation and the adaptation of microorganisms to changing environmental conditions are directly related to a phenomenon called “quorum sensing” (QS). This term refers to a communication system among bacteria, that allows microorganisms to interact with each other by generating and distributing molecular signals called auto-inductors. The concentration of these signals increases with increasing bacterial cell density [124]. Such auto-inductors can be the small oligopeptides in Gram-positive bacteria, whereas Gram-negative bacteria produce acyl homoserine lactones [124]. They attach to extracellular domains of the membrane histidine kinase receptor, causing autophosphorylation and cytoplasmic reaction. QS makes it possible for populations to coordinate their activity and modulate gene expression in response. Thanks to QS-dependent gene expression, the ability to maintain a “social” structure is ensured. This structure controls some basic physiological pathways and generates joint reactions, such as biofilm formation, pathogenesis, biodegradation of pollutants, mobility, sporulation, and virulence [124]. It is believed that QS participates in the production of virulence factors (pyoverdine, pyocyanin, rhamnolipids protease, alginate, and elastase of *P. aeruginosa*, lipase, protein A, fibronectin, enterotoxin, and hemolysins of *S. aureus*), as well as in the expression of antibiotic resistance genes [124]. Therefore, a disruption of microbial communication may contribute to a reduction in the pathogenic activity of bacteria. The development of quorum-sensing inhibition strategies could open up a new era in the treatment of infectious diseases. In this context, nanoparticles could become an excellent alternative to traditional antibiotics due to their effective delivery properties and better penetration ability.

An important strategy for suppressing QS is, therefore, inhibition of QS-mediated virulence factors in bacterial pathogens. The inhibition of virulence factors under the influence of “green” AgNPs was determined in *P. aeruginosa*, where AgNPs resulted in a dose-dependent effect on pyocyanin inhibition [115]. Pyocyanin is a blue redox-active secondary metabolite produced by *P. aeruginosa*. This pigment supports the development of biofilm and plays an overwhelming role in the body’s defense system [125]. Moreover, the pyocyanin pigment damages host cell membranes and contributes to the destruction of host tissues by generating ROS. The production of this pigment by *P. aeruginosa* was gradually decreased after treatment with AgNPs from *Lactobacillus rhamnosus* [125]. The production of another virulence factor, pyoverdin from *P. aeruginosa*, which plays a crucial role in host infection, and also causes iron deficiency in host tissues, competing for iron with mammalian transferrin, was suppressed under the influence of AgNPs [111]. Qais et al. suggest that inhibition of pyoverdin shows the protective effect of AgNPs synthesized by *M. koenigii* in reducing the virulence of *P. aeruginosa* [78].

Another pigment is a prodigiosin in *Serratia marcescens*, which is controlled by quorum sensing. It is believed that the loss of the hydrophobic dye alters the nature of the cell surface, ultimately leading to a weakening of the biofilm structure complex. This, in turn, promotes microcolonization and reduces EPS production. Treatment with AgNPs has been shown to inhibit prodigiosin synthesis. With increasing concentrations of *Moringa oleifera*-AgNPs, prodigiosin pigment synthesis was seen to decrease in a concentration-dependent manner [126].

Violacein is a natural bis–indole pigment whose production in *Chromobacterium violaceum* is regulated by the QS system depending on the density of the bacterial population. The *C. violaceum* bacterium can cause infections that can be dangerous to people with weakened immune systems. Violacein helps the bacteria bypass the immune system and increases their virulence [126]. And since QS inhibitors also suppress violacein production, biological compounds repressing its synthesis can also be considered potential therapeutic agents against bacterial infections. The inhibition of violacein synthesis was demonstrated for AgNPs synthesized by *Diospyros villosa* [127]. Pigment production was seen to be inhibited by 80.6% at 1/2× MIC (8 μg/mL) of *M. oleifera*-mediated AgNPs [126]. It was found that *Trachyspermum ammi*-NPs could significantly show an impact on violacein production at very low concentrations, 97.44% of violacein inhibition was recorded at 0.1 μg/mL [128]. It is known that the QS mechanism regulation in *C. violaceum* is influenced by the signaling molecule acylhomoserinlactone (AHL). Gram-negative bacteria transmit QS signals through N-acylhomoserine lactones (AHL), produced by AHL synthases (LuxI proteins). The structure of AHL consists of two parts: a homoserine lactone ring and a variable side chain. The length of this side chain determines whether AHL molecules are short-chain (C4–C8) or long-chain AHL (C10–C18). AHL diffuses from the cell into the extracellular environment. When a certain threshold is reached, they return to the cell and bind to the LuxR receptor, forming the LuxR/AHL complex and activating the expression of target genes, including genes encoding virulence factors. The most significant violacein inhibition was observed using a short-chain AHL biosensor under the influence of AgNPs derived from fruits, whereas AgNPs from leaves demonstrated the highest violacein inhibition using a long-chain AHL biosensor. The GC-MS analysis of AHL signaling molecules did not reveal significant peaks of specific AHL peaks under the influence of AgNPs from *Trachyspermum ammi* [128].

Another sign of pathogenic bacterial virulence is hemolysin production, which causes red blood cell lysis and leads to increased pathogenicity. β-hemolysis in *C. violaceum* under the influence of biosynthesized AgNPs at sub-inhibitory concentrations, i.e., 0.5 μg/mL and 1 μg/mL, was not observed [129].

The Las protein expression plays a crucial role in biofilm formation and regulated QS. *P. aeruginosa* produces several hydrolytic enzymes, such as elastase, which are responsible for tissue component destruction and, ultimately, disrupt the mechanisms involved in the host immune system. The dose-dependent effect of AgNPs from *C. copticum* and *L. rhamnosus* was shown on the elastase activity in *P. aeruginosa* [114,125].

Proteases are able to overcome the body’s defense system by breaking down proteins in host cells, leading to increased bacterial invasion. The production of exoproteases under the influence of AgNPs was suppressed in both *S. marcescens* and *P. aeruginosa* cells [114]. *Pteris vittata*-mediated AgNPs suppressed the production of virulence factors and limited QS. The activity of toxin protease and pyocyanin synthesis in *P. aeruginosa* cells treated with AgNPs was suppressed by 88% and 94%, respectively, depending on the concentration, when the cells were treated with 25 μg/mL of AgNPs [130].

Swimming motility is the rotation of flagella to propel a cell through a liquid medium. Such bacterial motility is extremely necessary for biofilm formation, its propagation, chemotaxis, and virulence, as well as in the regulation of QS signaling pathways. AgNPs could inhibit the swimming motility of *P. aeruginosa* and *S. marcescens* [114]. Similar data on the inhibition of swimming motility in *P. aeruginosa* and *S. marcescens* were obtained for AgNPs produced by *L. rhamnosus* [125].

Rhamnolipids play an important role in the attachment of bacterial cells to hard surfaces and the maintenance of the biofilm structure. Their production in *P. aeruginosa* is regulated by the RhlR-RhlI QS system, and it has been found that their synthesis decreases after AgNP treatment [127]. In a study involving AgNP-treated *M. koenigii*, a decrease in rhamnolipid production was observed. The presence of 1, 2, 4, and 8 μg mL^−1^ of AgNPs decreased the rhamnolipid production by 18.12, 36.55, 49.05, and 58.29% in the supernatant of *P. aeruginosa* PAO1, respectively [114].

Interesting data were received owing to molecular docking. It has been shown that the interaction and binding of Ag-NPs with QS-controlled proteins—autoinductor synthases (LasI and RhlI)—as well as with QS-controlled transcription regulatory proteins (LasR and RhlR) occurs due to electrostatic interaction with these proteins. Computer analysis showed that the LasI/RhlI synthase inhibition by AgNPs can block the AHL biosynthesis, as a result of which AHL is not produced and QS does not occur. This interference with regulatory transcription factors leads to the LasR/RhlR system inactivation, ultimately blocking the expression of virulence genes regulated by QS [130].

The conducted genetic studies (KEGG pathway enrichment analysis) indicated that AgNPs significantly slowed down the overall metabolism of *E. coli* cells: 54 downregulated genes, including substance modification and catabolic processes, energy production and transmission, and others, and quorum-sensing 16 downregulated gene pathways were observed [131]. Among them, *prpR*, *csiD*, *LSRa*, *ugpB*, *OSM*, *ydcJ*, *ytiD*, *prpR*, and *csiD* are the most suppressed. These are mainly canonical genes involved in the regulation of propionate catabolism transcription, which catalyzes glutarate (GA) and L-2-hydroxyglutarate hydroxylation. On the one side, the biosynthesis of the aminoacyl-tRNA synthetases and cofactors and the metabolism of thiamine and biotin were upregulated in 164 gene expressions. Four genes, namely *bioA*, *thiF*, *thiE*, and *rhlE*, exhibited the most significant upregulation in expression. The first three of these genes are transferases playing a crucial role in catalytic thiamine and biotin biosynthesis. Thiamine and biotin are two essential vitamins that are necessary for the conversion of certain nutrients into energy. RhlE, on the other hand, is an RNA helicase involved in ribosome assembly and improves ribosome biogenesis [131]. Gene expression analysis, experimental evolution in response to sublethal or bactericidal AgNP treatments, and gene editing reveal that bacteria acquire resistance mainly through two-component regulatory systems (CusS/CusR, ArcA/ArcB, and EnvZ/OmpR, functioning in AgNP resistance), especially those involved in metal detoxification, osmoregulation, and energy metabolism [131].

Of particular interest is the study of the cumulative effect of the nanoparticles themselves and the capping biomolecules on their surface. It is known that certain plant compounds have an anti-QS effect. For example, they can inhibit violacein synthesis. For instance, cis-cis-p-menthenolide extracted and isolated from a plant endemic to occidental Mediterranean Sea islands, *Mentha suaveolens* ssp. *insularis* [132]. Essential oils of various medicinal plants, phlorotannins from seaweed *Hizikia fusiforme*, eugenol, one of the active phenolic phytochemical compounds in *Piper betle* leaves, can be attributed to such compounds. The molecular docking of AgNPs conjugated with eugenol showed their significant interaction with QS regulatory proteins (LasR, LasI and Vfr). This research can help to develop nanoparticles with antibacterial properties, particularly against antibiotic-resistant bacteria capable of QS.

## 4. Toxicity

The toxicity assessment of AgNPs is a crucial step in understanding their properties and potential use before their introduction into medical practice. It is necessary to evaluate the risks of AgNP application as medicines, as well as to determine the extent and benefits to which they may pose a threat to human health.

### 4.1. In Vitro

Numerous studies have been conducted using nanotechnology platforms to investigate the diverse effects of AgNPs on human cells in vitro. These studies aim to provide a better understanding of the potential risks and benefits associated with the use of AgNPs in medicine. As a rule, this type of study is conducted alongside the analysis of the anti-cancer properties, and also holds great promise in the fight against this deadly disease. Research of the AgNP cytotoxicity revealed depending on factors such as size, shape, and the capping agents used in their formation. For monocytes, macrophages, and epithelial cells in the lungs, smaller particles were shown to be more toxic than larger particles. Spherical nanoparticles were also more toxic than cubic or prismatic nanoparticles. Additionally, capping agents can often play an important role, increasing or conversely reducing the toxic effect.

Fibroblasts are one of the most commonly used cells for assessing the toxicity of AgNPs. It was found that increasing the nanoparticle concentration derived from *Rosa santana* does not significantly affect the viability of a normal mouse fibroblast cell line (L929). These cells did not show significant toxicity at concentrations ranging from 0.1 to 5 μg/mL [133]. *Rumex vesicarius*-AgNPs demonstrated no toxicity to the L-929 cell line [75]. The cytotoxic effect of nanoparticles from *Allium ampeloprasum* was significantly lower for fibroblast-normal cells than on Hela cell line cancer cells. The authors suggest that biosynthesized AgNPs contain secondary metabolites that not only prevent AgNP agglomeration, but also increase their medicinal value [134]. *Astragalus spinosus*-AgNPs were cytotoxic against cancerous cell lines while being nontoxic for human normal oral fibroblasts (NOF18) [96]. AgNPs synthesized using black tea extracts exhibit lower cytotoxicity against normal human primary fibroblasts and high toxicity toward ovarian carcinoma cells [135]. AgNPs showed no inhibition against the L-929 cell lines at lower concentrations, and plant extract did not show any toxicity against the L-929 cell line and proved its potential use in the synthesis of AgNPs [52]. AgNPs synthesized from *Pseudomonas chrysosporium* showed that up to a concentration of 12.5 μg/mL, they did not have significant toxicity toward mouse embryo fibroblast cells [136].

Human blood cell lines are also used to assess the toxic effect. Thus, AgNPs reduced the viability of White blood cells (WBCs) in a dose-dependent manner [137]. The AgNP solution exhibited lower cytotoxicity, depending on the dose and time, against peripheral blood mononuclear cells (PBMC) of the immune system consisting of lymphocytes (T cells, B cells, NK cells) and monocytes. This cytotoxicity of the PBMC indicates the potential for immune reactions and immunotoxicity, allowing for the prediction of immune effects during preclinical testing, as well as helping to understand the interaction between nanomaterials and immune system cells [138]. Concentrations of AgNPs (10 μM and 50 μM) from belladonna mother tincture in relation to PBMC cells do not show toxicity, because cell viability at both dosages is more than 75% [43]. Experiments conducted on human peripheral blood lymphocytes (HPBL) treated with 1.3 μg/mL of biogenic AgNPs for 24 h showed that nanoparticles did not harm the body’s first line of immune defense [139]. The biocompatibility of nanoparticles has been demonstrated with human red blood cells. Green AgNPs did not cause hemolysis of erythrocytes up to a concentration of 150 μg/mL, and in fact, the nanoparticles had a stabilizing effect on the cell membrane due to their lower hemolytic activity compared to the negative controls [140]. The hemolysis activity increased with increasing concentrations of AgNPs, a hemolysis is size dependent—the small-size nanoparticles (15 nm) showed higher hemolysis activity ~60% than the larger-size particles [50]. The low toxicity level of 8 nm green-synthesized AgNPs of average size against human erythrocytes belonging to the ‘O’ positive blood group indicates their safety for further potential use in pharmacological applications [96]. The analysis of the effects of bionanoparticles and chemically produced AgNPs on macrophages—immune cells that play a significant role in the body’s nonspecific immunity and the inflammatory process—suggests that the former is more beneficial, as their cytotoxic effects were reduced, which protected the immune ability of inflammatory response in mice. An important indicator, among other things, is that AgNPs have anticoagulant and thrombolytic activities at low concentrations, by maintaining a healthy blood condition and preventing the aggregation of blood cells, which contributes to the formation of blood clots [141].

Despite a lot of studies indicating that AgNPs do not have a toxic effect on human cells, there is still evidence to suggest otherwise. AgNPs have been found to accumulate in various organs and tissues, such as the liver and kidneys, where they may contribute to oxidative stress, DNA damage, and inflammation. These effects can lead to various diseases. Nanoparticle toxicity is closely linked to their capacity to generate free radicals, which can damage cellular molecules, leading to the loss of cell function, necrosis, apoptosis, and cell death. For instance, *Saccharomyces cerevisiae*-AgNPs were shown to be more toxic to normal keratinocytes and fibroblasts than to cancer cells. The toxic effect of AgNPs from *Leucophyllum* frutescens and *Russelia equisetiformis* on normal breast epithelial cells was stronger than on colorectal cancer cells. This NP toxicity with respect to normal cell lines is the main reason for limiting their medical use in cancer treatment and antibacterial therapy [142]. The direct interaction of nanoparticles with erythrocytes can damage their membranes, leading to their rupture or hemolysis. The degree of erythrocyte hemolysis depends on the size and concentration of the nanoparticles, with smaller AgNPs causing more severe hemolysis than larger AgNPs. The toxic effect of AgNPs derived from *Streptococcus aizuneis* on normal lung and liver cells showed a dose-dependent response, with the greatest decrease in cell viability achieved at a concentration of 0.2 ppm AgNPs [143]. The green nanoparticles were found to be toxic to both Vero cells and HeLa cancer cells, and even greater cytotoxicity was observed in reference human fibroblasts (WS-1) [144]. In experiments evaluating the toxicity of AgNPs by exposing *Cyprinus carpio* to various concentrations, it was found that nanoparticles accumulate in the intestines, gills, liver, muscles, kidneys, and brains of fish. This bioaccumulation promotes histological changes and destruction of intestinal villi, liver cell regeneration, and degeneration of the gill plate [145]. A similar study of fish cell lines also found that AgNPs cause various morphological changes in spindle-shaped fibroblast cells from three fish species (*Oreochromis niloticus* liver cells, *Cyprinus carpio* koi fin cells, and *Cyprinus carpio* gill). AgNPs had a pronounced cytotoxic effect, including cell contraction, rounding, and cell fusion in all three cell lines. The authors of the study suggest that this effect is due to the presence of biologically active compounds acting as capping agents during the green synthesis of AgNPs [146].

As mentioned above, one of the main challenges to the full-fledged potential use of AgNPs in medical applications is their ability to accumulate in different organs and tissues. However, biologically synthesized nanoparticles that bear a variety of biomolecules on their surface, which contribute to reducing toxicity and enhancing antibacterial efficacy, have the potential to have a more positive effect than those obtained through physicochemical methods. Numerous experiments in this field have been conducted both in vitro and in vivo. As a result, increased cytobiocompatibility of green-synthesized AgNPs with human mesenchymal stem cells (hMSC) and the 293T cell line can be achieved thanks to the presence of biologically active and biocompatible secondary metabolites from leaf extract [147].

### 4.2. In Vivo

Moreover, it was found that *Rhizophora apiculata* AgNPs are effective in protecting Swiss albino male mouse livers from damage provoked by carbon tetrachloride [148]. AgNPs from the ethanol extract of *T. cordifolia* leaves have demonstrated antioxidant, hepatoprotective, and nephroprotective effects. A significant recovery of serum levels of urea, creatinine, and uric acid, as well as SGOT and SGPT, in Swiss albino mice induced by potassium bromate, was observed after treatment with AgNPs, returning these levels to normal [149]. AgNPs synthesized by *Acacia rigidula* did not show any adverse or toxic effects on the nephrons and kidney function of treated female adult Wistar rats. The kidneys regulate the volume (amount of fluid in the body) and glucose concentration (associated with tubule function in nephrons), as well as the total protein concentration and creatinine concentration in both plasma and urine (associated with glomerular filtration in nephrons). These parameters were maintained at control levels in the animals [150]. The concentration of alanine aminotransferase (ALT) and aspartate aminotransferase (AST) (biomarkers of hepatic cell necrosis) and the concentration of albumin (biochemical biomarkers of damage and hepatic function) of the treated groups with AgNPs were similar to the values of control groups, moreover, pretreatment of AgNPs prevented liver cell damage caused by paracetamol at 100 and 200 mg/kg bw doses. In vivo analysis showed that treatment with *Litsea cubeba*-AgNPs contributed to the healing of wounds infected with MRSA. After 14 days of treatment, the wounds of the mice treated with *L. cubeba*-AgNPs were already healed, while the wounds without any treatment had not healed [83].

## 5. Conclusions

Using the biological potential of living organisms, such as bacteria, fungi, and plants, with modern nanotechnology in the form of environmentally friendly AgNPs, can open a new era of treatment approaches for various human diseases. Despite AgNP biosynthesis having several challenges, including difficulty in precisely regulating the shape and size of the biosynthesized AgNP majority, which affects their physicochemical properties, an eco-friendly approach to replacing hazardous compounds with non-toxic biomolecules during the biosynthesis process represents an extremely promising trend. A wide range of data supports AgNP application as an effective tool for antibacterial, anti-biofilm, and anti-quorum sensing activities. The multiplicity of processes influenced by nanoparticles on bacterial cells includes both cellular and molecular mechanisms, such as adhesion, penetration into the cell, interaction with cellular components and macromolecules, and ROS generation, and following this, bacterial cell death occurs. Moreover, AgNPs can interact not only with cellular components but also with the biofilm components, whose formation by antibiotic-resistant microorganisms greatly complicates the fight against pathogens. While antibiotics are unable to penetrate the matrix of biofilms, the small size of AgNPs enables particles to interact with bacterial cells, thus preventing biofilm formation. Moreover, it appears that AgNPs exhibit a stronger antibiofilm effect compared to nanoparticles of other metals, such as gold, as demonstrated in several comparative studies. AgNPs not only destroy biofilms but also prevent their formation. For example, AgNP-impregnated cotton and polyester fabrics were shown to suppress the growth of *S. aureus* and *E. coli* by 100%. Such functionalized fabrics can be used to develop hospital clothing and medical devices to prevent microorganism transmission and reduce the risk of hospital-acquired infections. Additionally, the AgNP ability to disrupt the QS cascade by suppressing signaling molecules and blocking receptors suggests their broad antibacterial properties.

Besides the factors such as the shape, size and dose of AgNPs contributing to their antibacterial effect, capping agents seem to play a significant role in enhancing the anti-pathogenic properties. The most intriguing biological compounds found in plants, fungi, and bacterial cells possess such properties. The medicinal properties, primarily from plants, have been known to humans since ancient times and are widely used in traditional medicine, such as Chinese traditional medicine, Ayurveda, and alternative medicine around the world. The identification of these biomolecules, combined with modern nanotechnology techniques, could create a new medicine generation. Polysaccharides, proteins, enzymes, polyphenols, alkaloids, anthraquinones, flavonoids, terpenoids, glycosides, and steroids, acting as capping agents, enhance the beneficial properties of AgNPs. Knowing these metabolites allows for the creation of nanoparticles with specific therapeutic properties. A significant number of experiments where synthesis was made using individual bioactive compounds rather than extracts has confirmed this. For instance, AgNPs functionalized with N-acylated homoserine lactones and lactonase exhibited a substantial reduction in the biosynthesis of exopolysaccharides, metabolic activity, and hydrophobicity of the bacterial cell surface, as well as an inhibitory effect on biofilm development in multidrug-resistant *K. pneumoniae* strains. The application of hesperidin and pectin in the synthesis process revealed that these AgNPs induce damage to the *E. coli* cell wall, resulting in internal component leakage and a subsequent increase in ROS. AgNPs in combination with pterostilbene, a plan-derived compound and a derivative of trans-stilbene that is mainly found in blueberries and grapes, have demonstrated an excellent antibacterial effect against *S. aureus*. Molecular docking has revealed the potential affinity of compounds (certain polyphenols) present in *Valeriana jatamansi*-AgNPs for the target protein of *Str. mutans*, which contributes to their antimicrobial activity [123].

The potential benefits of AgNPs must be considered in light of their toxicity. It is crucial to strike a balance between the necessary dose and the toxicity of AgNPs. While many studies have shown that AgNPs are biocompatible with human and animal cells, there is also evidence of their high toxicity. AgNP accumulation in vital human organs and its effects on various tissues, intolerance to silver, allergic reactions, and other negative consequences, as well as the tendency to form aggregates, which can increase their potential toxicity, need to be thoroughly investigated before they are implemented in medical practice. Despite this, AgNPs remain extremely attractive due to their potential use in combating pathogenic microorganisms. Biological factories could produce biocompatible, biologically active, low-toxic nanoparticles with a wide range of beneficial properties.

## Figures and Tables

**Figure 1 antibiotics-14-00005-f001:**
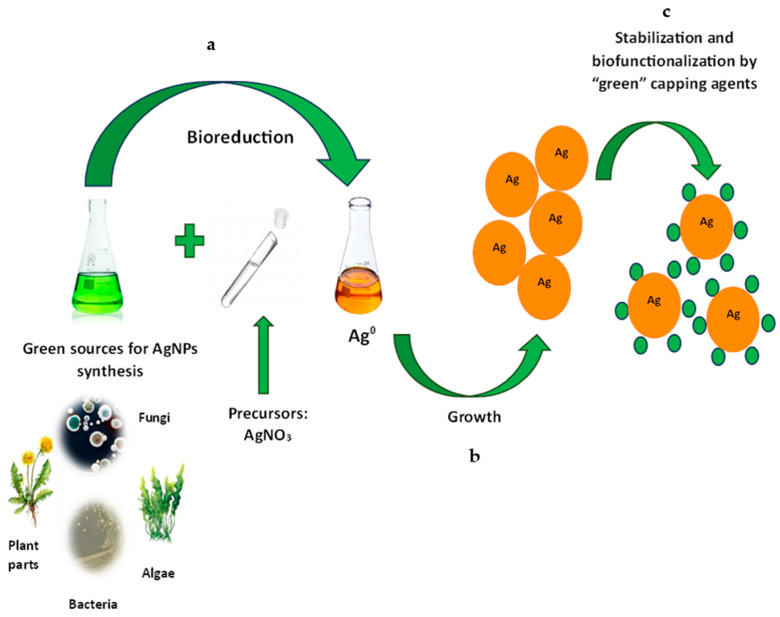
Biosynthesis of AgNPs. (**a**) reduction by different biological sources; (**b**) nanoparticle’s growth; (**c**) stabilization and capping by plant, fungal or bacterial compounds.

**Figure 2 antibiotics-14-00005-f002:**
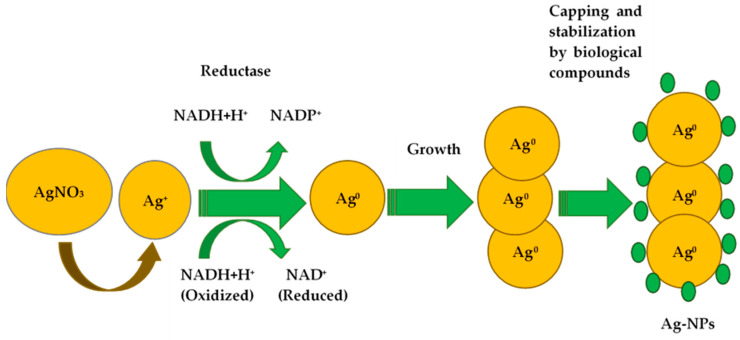
The biosynthesis of AgNPs in bacteria with the participation of NADH-dependent reductase.

**Figure 3 antibiotics-14-00005-f003:**
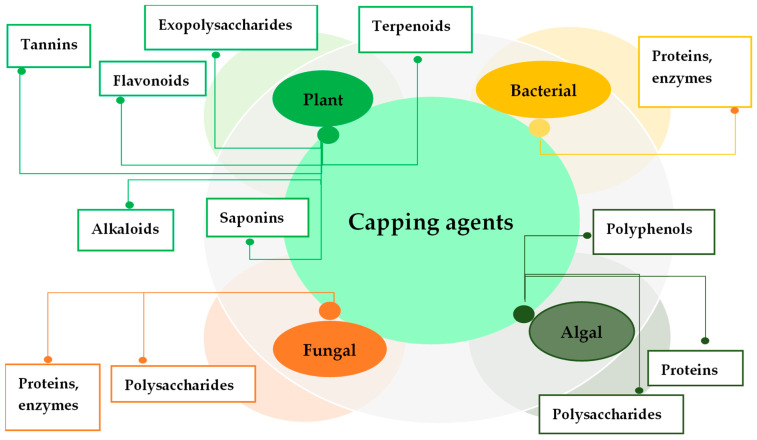
Capping agents from different biological sources.

**Figure 4 antibiotics-14-00005-f004:**
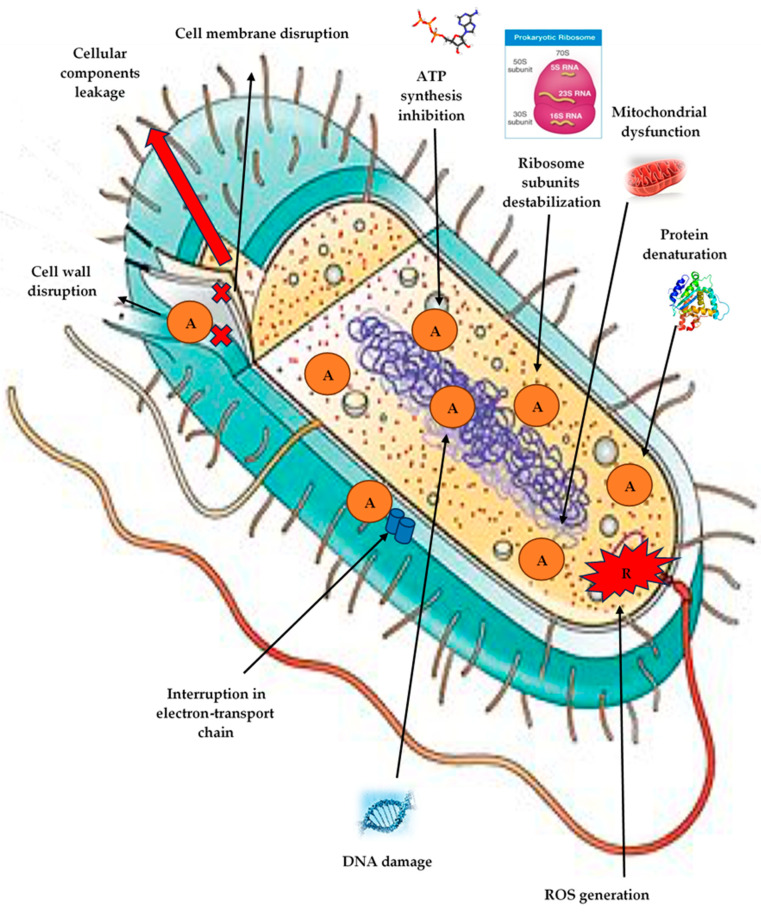
The proposal mechanism of AgNP antibacterial activity.

**Figure 5 antibiotics-14-00005-f005:**
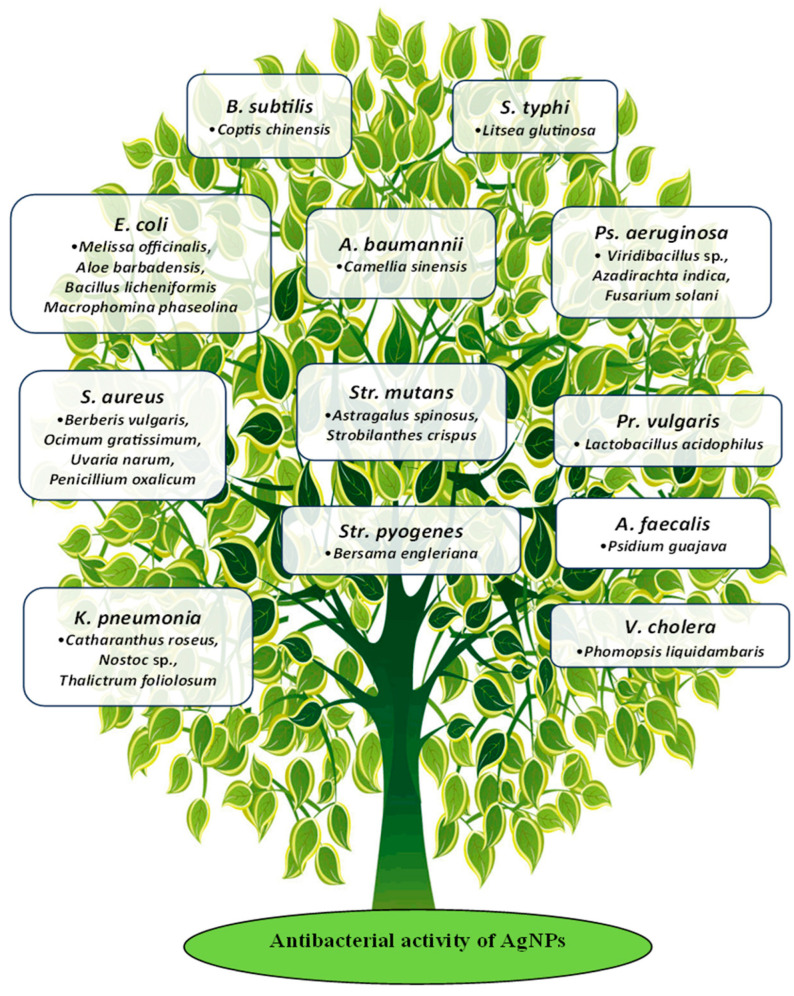
Antibacterial activity of AgNPs. The AgNPs source is indicated by a dot (•).

**Table 1 antibiotics-14-00005-t001:** Bacteria-mediated AgNPs.

Microorganism	Type of Synthesis	Shape	Size, nm	Ref.
*K. rhizophila*	Extracellular	spherical	10–200	[4]
*Planomicrobium* sp.	Extracellular	spherical	1–10	[5]
*Exiguobacterium aurantiacumm*	Extracellular	spherical	5–50	[6]
*Lactobacillus brevis*	Extracellular	spherical, polyhedral	5–40	[7]
*Bacillus* sp.	Extracellular	spherical	10–60	[8]
*Enterococcus* sp.	Extracellular	spherical	10–80	[10]
*A. faecalis*	Extracellular	spherical	30–50	[10]
*Pseudomonas aeruginosa*	Extracellular	spherical	~11	[11]
*Enterobacter cloacae*	Intracellular	spherical	7–25	[3]
*Cupriavidus necator*, *B. megaterium*, and *B. subtilis*	Extra- and Intracellular	spherical	20.8–118.4	[13]

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
