# Peer review of "Green Silver Nanoparticles: An Antibacterial Mechanism"

_antibiotics, 2024, doi:10.3390/antibiotics14010005_

Round 1

Reviewer 1 Report

Comments and Suggestions for Authors

Summary: The review focuses on the synthesis of silver nanoparticles by biological methods, with emphasis on the underlying mechanisms and governing parameters for controlling the size and shape of nanoparticles.

The review topic is likely to attract the attention of the Antibiotics audience. The scope of work covered is appropriate. However, the manuscript should be revised to reach an acceptable level of accuracy, precision, and clarity before it can be accepted for publication. One recurring issue is that the manuscript uses qualitative terms in places where quantitative descriptions would be more appropriate and informative. Furthermore, the manuscript is plagued with  a casual non-academic tone in several places, spelling mistakes, and typos.

A partial list follows below, but it is suggested that the authors take a critical approach to revising and proofreading their manuscript with this feedback in mind.

Recommendation: Reconsider after major revisions.

Comments:

·          Page 2, Line 46-47: “Green silver nanoparticles were destined to become the most famous.” There are several metals which have reported antibacterial properties for e.g., copper, zinc, gold etc. The author’s statement hints that silver NPs have superior antibacterial properties compared to others, which is misleading to the reader.

It is suggested to reword the sentence and use a more academic tone while rewriting.

·       Page 2, Line 55-56: “The number of publications since the 2000s was measured in thousands, if not tens of thousands.” Please provide references for some seminal reports to support this statement.

·       Page 2, Line 87-88: “The scheme for obtaining silver nanoparticles using "biofactories" is quite simple.”

The author discusses in the manuscript the influence of key parameters like pH, salt concentration, temperature and incubation time affecting the nano particle synthesis. Thus, it is misleading to mention that the Ag NP synthesis is quite simple. It is recommended to rewrite the sentence to better reflect the information provided in the manuscript.

·       Page 2, Line 89-90: “An important difference from physical and chemical methods is the…” Please add a brief description of the physical and chemical methods before highlighting the differences between them, for clarity to reader.

·       Page 2, Line 90-91: “various phytocompounds, et. in the nanoparticle reduction and stabilization” Please mention the specific phytocompounds responsible for reduction and stabilization.

·       Page 2, Line 93: “Ag+ is reduced to Ag0 using biological catalysts in a "factory" of synthesis”. Please mention the biological catalysts which are responsible for this reduction reaction and provide relevant references.

Can the author clarify what does “factory of synthesis” denote? It is recommended to rewrite for better clarity to reader.

·       Page 2, Line 95: “third, termination occurs, leading to the formation of final form”. The author mentions the 3 stages for NP synthesis: reduction, cluster formation followed by stabilization, and termination.

What process is the termination step terminating? As the final form of the NP is obtained by stabilization of the clusters, can “stabilization” be considered the final stage and not “termination’?

·       Page 3, Line 96-97: “Functional groups in biomolecules are capable of releasing electrons in the reaction medium.” The statement is heavily generalized and does not provide sufficient relevant information for the reader.

What are the functional groups? Which are the biomolecules being mentioned in this statement? What is the solvent/reaction medium in this reaction? It is recommended to rewrite while trying to incorporate these details for being more informative to reader.

·       Page 3, Line 98: “As a result of the reduction, particle formation begins”. Can the author please elaborate on why reduction is necessary for particle formation and cite relevant articles if needed.

·       Page 3, Line 104: “Capping agents interact with Ag0..” It is suggested to elaborate on the chemistry of these capping agents. Specifically, their chemical names, functional groups, and plausible mechanism for action. As the author mentions the crucial role of caping agents in limiting the size of the NPs, we believe a thorough discussion on their chemistry will be informative to the reader.

The author provides more discussion on capping agents in Figure 2, Page 12. However, it gives an overview of the different types of capping agents but there is little discussion on their chemistry.

·       Page 3, Line 104: Please change “Ag0” to “Ag0” to maintain uniformity with Figure 1.

·       Page 3, Line 111: Please change “pH solution” to “pH of solution”.

·       Page 3, Line 112-113: “However, these parameters should be selected primarily based on the biological objects capable of synthesizing AgNPs.” Please rewrite the sentence to convey information for better clarity.

·       Page 3, Line 116: Please change the title for Figure 1. The different steps involved in NPs synthesis like reduction, growth, and stabilization can be included in Figure title using 1(a), (b), and (c) format.

The image used in Figure 1 is blurry and not high-resolution. Please change this and also the subsequent figures in manuscript.

What is the source of images in Figure 1? If it is taken from an existing article, please add relevant reference and copyright permission.

·       Page 3, Line 121-122: “The undeniable advantage of extracellular synthesis is the simplicity extracting the silver nanoparticles from the solution.”

Please elucidate the difference between extra and intracellular NP synthesis and why extracellular is simple, before concluding the undeniable advantage.

·       Page 4, Line 133: “membrane proteins transporting the silver ions into cells”. Please specify which membrane proteins are responsible for the transport of Ag ions.

·       Page 4, Line 143: “The main enzymes involved are thought to be nitrate reductases..”

It is advised to use academic language for the sentences. “are thought to be” gives the impression that this statement is debatable and lacks certainty.

If the author believes that there are conflicting literature reports on the identity of the main enzyme, it is recommended to add a brief discussion on these reports and conclude that establishing the nature of the main enzyme is an active point of interest in the community.

·       Page 4, Line 144: Please change “bioreduction” to “reduction”. Reduction is the chemical reaction that the biological enzymes are catalyzing. However, “bioreduction” can be mis-interpreted by the reader as reducing the number of microorganisms in a system.

·       Page 4, Line 150: Please change “nanosilver” to “nanoparticles of silver”.

·       Page 4, Line 148-149: “The enzyme receives electrons from NADH and oxidizes them to NAD+, and then undergoes oxidation, reducing silver ions into nanosilver.”

It is evident that a major focus of the manuscript is to elucidate the synthesis if Ag NPs by various biological systems and its underlying mechanisms. Thus, it is recommended to present the crucial mechanism related information in the form of a figure, to bring it to the focus of reader.

It is advised to indicate the redox reaction being mentioned in the sentence in Figure 1 (if possible) or create an additional figure to highlight the different reaction mechanisms being discussed in the manuscript.

·       Page 4, Line 153: Please change “secret” to “secrete”.

·       Page 4, Line 152-153: “Lactobacillus species have the ability to secret an aldehydic group through additional polysaccharides, which are responsible for the reduction of Ag+ to Ag0”.

The sentence is unclear and uses a non-academic language. An aldehydic group cannot be secreted, rather a compound with aldehyde functionality is secreted by Lactobacillus. Please specify the compound which is responsible for the reduction?

·       Page 4, Line 157: Please change “hypothetical mechanism” to “plausible mechanism”.

·       Page 4, Line 160: “Ag ions”. Please mention the oxidation state of the Ag ions in the sentence.

·       Page 4, Line 163: Please change “initiation synthesis of AgNPs” to “initiate the synthesis of AgNPs”.

·       Page 5, Line 167: “Ag (I) to Ag (0)”. There are 3 different variations of oxidation state in the manuscript, e.g., Ag0, Ag0, and Ag (0). Please use uniform format to indicate the oxidation state.

·       Page 5, Line 209-211: Can the author add a brief description on how the shape (e.g., spherical, or triangular) of AgNPs changes based on the reaction rate (e.g., high or low)? Please add relevant references as well.

·       Page 5, Line 213: Please italicize “B. cereus”.

·       Page 6, Line 226: Please change “AgNPs.” to “AgNPs”.

·       Page 6, Line 226: Please change “the formation larger” to “the formation of larger”.

·       Page 6, Line 229: What is LB media used as an acronym for? Please mention.

·       Page 6, Line 231: “maximum synthesis was observed”. Can the author comment on what does maximum synthesis denote? Please specify the percentage/ratio reported in the literature, which indicates the greater amount of product obtained in the stationary phase.

·       Page 13, Figure 3: What is the source of Figure 3? If it is taken from a published article, please add citation and copyright permission.

Please consider this comment for all the figures in the manuscript.

·       References:

Please correct the format for references 4, 6, 162, 222, 227, 228, 232, 243, 244, 263.

Please add journal name, volume, page no. for reference 223.

Comments on the Quality of English Language

The manuscript uses an overly casual tone in a number of places, which compromises the academic language of the manuscript. Further, there is frequent use of uncommon and non-academic keywords (e.g. nanosilver), and typo mistakes are present throughout the manuscript. Thus detailed proofreading is recommended, with an emphasis on using an academic style of writing. 

Author Response

Dear colleague,

Thank you very much for your comprehensive analysis of our work!

Appropriate changes were made in accordance with your comments. The changes are highlighted in green.

Comment 1: Page 2, Line 46-47: “Green silver nanoparticles were destined to become the most famous.” There are several metals which have reported antibacterial properties for e.g., copper, zinc, gold etc. The author’s statement hints that silver NPs have superior antibacterial properties compared to others, which is misleading to the reader.

It is suggested to reword the sentence and use a more academic tone while rewriting.

Answer: This sentence was rewritten.

Comment 2: Page 2, Line 55-56: “The number of publications since the 2000s was measured in thousands, if not tens of thousands.” Please provide references for some seminal reports to support this statement.

Answer: This sentence was rewritten. The Reference 6 support this statement.

Comment 3: Page 2, Line 87-88: “The scheme for obtaining silver nanoparticles using "biofactories" is quite simple.”

The author discusses in the manuscript the influence of key parameters like pH, salt concentration, temperature and incubation time affecting the nano particle synthesis. Thus, it is misleading to mention that the Ag NP synthesis is quite simple.

Answer: This sentence was rewritten.

Comment 4: Page 2, Line 89-90: “An important difference from physical and chemical methods is the…” Please add a brief description of the physical and chemical methods before highlighting the differences between them, for clarity to reader.

Answer: The information about chemical and physical methods was added to highlight the differences between them and biological methods. The reference was also added.

Comment 5: Page 2, Line 90-91: “various phytocompounds, et. in the nanoparticle reduction and stabilization” Please mention the specific phytocompounds responsible for reduction and stabilization.

Answer: Examples of phytocompounds were added. The participation of these compounds is also discussed in detail later in the text.

Comment 6: Page 2, Line 93: “Ag+ is reduced to Ag0 using biological catalysts in a "factory" of synthesis”. Please mention the biological catalysts which are responsible for this reduction reaction and provide relevant references.

Can the author clarify what does “factory of synthesis” denote? It is recommended to rewrite for better clarity to reader.

Answer: This sentence was rewritten for better clarity to reader. The definition of the term "synthesis factory" was given. Reference also added.

Comment 7: Page 2, Line 95: “third, termination occurs, leading to the formation of final form”. The author mentions the 3 stages for NP synthesis: reduction, cluster formation followed by stabilization, and termination.

What process is the termination step terminating? As the final form of the NP is obtained by stabilization of the clusters, can “stabilization” be considered the final stage and not “termination’?

Answer: This sentence was rewritten for better clarity to reader. Nanoparticles synthesized in extracts of plants, fungi, bacteria have a functionalized surface that can contain various proteins, enzymes polysaccharides, polyatomic alcohols, tannins etc. The presence of these biological can stabilize silver nanoparticles, prevent agglomeration and create their capping in an aqueous solution.

Comment 8: Page 3, Line 96-97: “Functional groups in biomolecules are capable of releasing electrons in the reaction medium.” The statement is heavily generalized and does not provide sufficient relevant information for the reader.

What are the functional groups? Which are the biomolecules being mentioned in this statement? What is the solvent/reaction medium in this reaction? It is recommended to rewrite while trying to incorporate these details for being more informative to reader.

Answer: This sentence was rewritten. This section describes a brief scheme of AgNPs biosynthesis. More detailed information about the biological compounds involved in this process is provided in the sections describing biological sources - bacteria, fungi, plants. For example, Nicotinamide adenine dinucleotide (NAD) and NADH-dependent enzymes are the basis for the biogenic synthesis of these nanomaterials. The reduction process is induced by electron transfer from NADH using NADH-dependent reductase as an electron transporter. Nitrate ions from the AgNO3 salt induce nitrate reductase. The enzyme receives electrons from NADH and oxidizes them to NAD+, and then undergoes oxidation, reducing silver ions into silver nanoparticles.

Comment 9: Page 3, Line 98: “As a result of the reduction, particle formation begins”. Can the author please elaborate on why reduction is necessary for particle formation and cite relevant articles if needed.

Answer: This sentence was rewritten for better clarity to reader. It means that the appearance of the color associated with the plasmon resonance band of nanoparticles was a qualitative indication of the formation of these particles.

Comment 10: Page 3, Line 104: “Capping agents interact with Ag0..” It is suggested to elaborate on the chemistry of these capping agents. Specifically, their chemical names, functional groups, and plausible mechanism for action. As the author mentions the crucial role of caping agents in limiting the size of the NPs, we believe a thorough discussion on their chemistry will be informative to the reader.

The author provides more discussion on capping agents in Figure 2, Page 12. However, it gives an overview of the different types of capping agents but there is little discussion on their chemistry.

Answer: This sentence was corrected and supplemented. More detailed information about capping agents is provided in the relevant section. Unfortunately, there is not much information about the molecular mechanisms of interaction of such compounds with silver nanoparticles. The main part of the data is obtained by FTIR analysis. The identification of compounds forming a biological "corona" is interesting primarily because these coating substances themselves exhibit antimicrobial action, resulting in a synergistic effect of metal nanoparticles and the coating itself. At the same time, it is known that protein components in the medium can bind to AgNPs through free amino or cysteine groups, exopolysaccharides contain negatively charged functional groups, such as am-ide, carboxyl, and hydroxyl hydroxyls, which are able to interact with metal cations, flavonoid amide groups have a strong affinity for metal ions etc. All this information is presented in the section "capping agents". Some information about the interaction of biological compounds with silver nanoparticles was also added and supported with references.

Comment 11: Page 3, Line 104: Please change “Ag0” to “Ag0” to maintain uniformity with Figure 1.

Answer: Ag0” was changed to “Ag0.

Comment 12: Page 3, Line 111: Please change “pH solution” to “pH of solution”.

Answer: “pH solution” was changed to “pH of solution”.

Comment 12: Page 3, Line 112-113: “However, these parameters should be selected primarily based on the biological objects capable of synthesizing AgNPs.” Please rewrite the sentence to convey information for better clarity.

Answer: This sentence was rewritten.

Comment 13: Page 3, Line 116: Please change the title for Figure 1. The different steps involved in NPs synthesis like reduction, growth, and stabilization can be included in Figure title using 1(a), (b), and (c) format.

The image used in Figure 1 is blurry and not high-resolution. Please change this and also the subsequent figures in manuscript.

What is the source of images in Figure 1? If it is taken from an existing article, please add relevant reference and copyright permission.

Answer: All images in this paper was made by author. The image quality was improved. The tittle was corrected.

Comment 14: Page 3, Line 121-122: “The undeniable advantage of extracellular synthesis is the simplicity extracting the silver nanoparticles from the solution.”

Please elucidate the difference between extra and intracellular NP synthesis and why extracellular is simple, before concluding the undeniable advantage.

Answer: This sentence was rewritten. The information about the difference between extracellular and intracellular synthesis was added. The extracellular synthesis method is preferable due to easy and simpler purification steps. In contrast, the intracellular NPs synthesis method is challenging and expensive due to the involvement of additional separation and purification processes.

Comment 15: Page 4, Line 133: “membrane proteins transporting the silver ions into cells”. Please specify which membrane proteins are responsible for the transport of Ag ions.

Answer: Unfortunately, information on the intracellular microbial synthesis of nanoparticles is extremely scarce, including on the biological compounds involved in this process. This sentence has been replaced with a more accurate and acceptable one. Relevant references were added.  

Comment 16: Page 4, Line 143: “The main enzymes involved are thought to be nitrate reductases..”

It is advised to use academic language for the sentences. “are thought to be” gives the impression that this statement is debatable and lacks certainty.

If the author believes that there are conflicting literature reports on the identity of the main enzyme, it is recommended to add a brief discussion on these reports and conclude that establishing the nature of the main enzyme is an active point of interest in the community.

Answer: This sentence was rewritten.

Comment 17: Page 4, Line 144: Please change “bioreduction” to “reduction”. Reduction is the chemical reaction that the biological enzymes are catalyzing. However, “bioreduction” can be mis-interpreted by the reader as reducing the number of microorganisms in a system.

Answer: “Bioreduction” was changed to “reduction”.

Comment 18: Page 4, Line 150: Please change “nanosilver” to “nanoparticles of silver”.

Answer: “Nanosilver” was changed to “silver nanoparticles”.

Comment 19: Page 4, Line 148-149: “The enzyme receives electrons from NADH and oxidizes them to NAD+, and then undergoes oxidation, reducing silver ions into nanosilver.”

It is evident that a major focus of the manuscript is to elucidate the synthesis if Ag NPs by various biological systems and its underlying mechanisms. Thus, it is recommended to present the crucial mechanism related information in the form of a figure, to bring it to the focus of reader.

It is advised to indicate the redox reaction being mentioned in the sentence in Figure 1 (if possible) or create an additional figure to highlight the different reaction mechanisms being discussed in the manuscript.

Answer: The Figure 2 was added to highlight bacterial mechanism of AgNPs synthesis. The figure was made by author.

Comment 20: Page 4, Line 153: Please change “secret” to “secrete”.

Answer: The word was changed.

Comment 21: Page 4, Line 152-153: “Lactobacillus species have the ability to secret an aldehydic group through additional polysaccharides, which are responsible for the reduction of Ag+ to Ag0”.

The sentence is unclear and uses a non-academic language. An aldehydic group cannot be secreted, rather a compound with aldehyde functionality is secreted by Lactobacillus. Please specify the compound which is responsible for the reduction?

Answer: This sentence was deleted.

Comment 22: Page 4, Line 157: Please change “hypothetical mechanism” to “plausible mechanism”.

Answer: “Hypothetical mechanism” was changed to “plausible mechanism”.

Comment 23: Page 4, Line 160: “Ag ions”. Please mention the oxidation state of the Ag ions in the sentence.

Answer: The oxidation state of the Ag ions was added.

Comment 24: Page 4, Line 163: Please change “initiation synthesis of AgNPs” to “initiate the synthesis of AgNPs”.

Answer: “Initiation synthesis of AgNPs” was changed to “initiate the synthesis of AgNPs”.

Comment 25: Page 5, Line 167: “Ag (I) to Ag (0)”. There are 3 different variations of oxidation state in the manuscript, e.g., Ag0, Ag0, and Ag (0). Please use uniform format to indicate the oxidation state.

Answer: The format of the oxidation state was uniformed.

Comment 26: Page 5, Line 209-211: Can the author add a brief description on how the shape (e.g., spherical, or triangular) of AgNPs changes based on the reaction rate (e.g., high or low)? Please add relevant references as well.

Answer: The reaction rate depends on factors such as temperature, pH, and precursor concentration. These factors, in turn, influence the size and shape of silver nanoparticles. For example, at high alkaline pH, spherical and rod-shaped AgNPs were created due to a high reaction rate, while triangular or polygonal AgNPs were synthesized at lower pH due to the low reaction rates. The size of B. cereus-silver nanoparticles decreased with increased pH, in alkaline pH, monodispersed, spherical nanoparticles formed with an increased amount. At high pH, the reaction rate was upgraded resulting in nucleation and growth of small-sized silver nanoparticles. At higher concentrations of AgNO3, the reduction rate of silver ions into AgNPs. increases, while a high concentrations of silver salt led to the formation larger aggregated nanoparticle. The use of microorganisms for the production of silver nanoparticles is less popular than the use of plants. The available information is provided.

Comment 27: Page 5, Line 213: Please italicize “B. cereus”.

Answer: It was changed.

Comment 28: Page 6, Line 226: Please change “AgNPs.” to “AgNPs”.

Answer: “AgNPs.” Was changed to “AgNPs”.

Comment 28: Page 6, Line 226: Please change “the formation larger” to “the formation of larger”.

Answer: “The formation larger” was changed to “the formation of larger”.

Comment 29: Page 6, Line 229: What is LB media used as an acronym for? Please mention.

Answer: Luria-Bertani (LB) is the most widely used medium for the growth of bacteria. A transcript was added.

Comment 30: Page 6, Line 231: “maximum synthesis was observed”. Can the author comment on what does maximum synthesis denote? Please specify the percentage/ratio reported in the literature, which indicates the greater amount of product obtained in the stationary phase.

Answer: This sentence was replaced with a more adequate one. Some information was added. The authors suggest that, since cells are under stress during the stationary phase, this can cause the production and secretion of small and diffusive compounds, such as exopolysaccharides, enzymes, and proteins, that are able to interact with insoluble metals and cause their reduction. It has been proposed that the extracellularly secreted enzyme nitrate reductase is responsible for the reduction of silver ions to produce silver nanoparticles. the LB media seemed to be more efficient for the production of nanoparticles with a uniform diameter than the other two media tested. Numeric values are not given.

Comment 31: Page 13, Figure 3: What is the source of Figure 3? If it is taken from a published article, please add citation and copyright permission.

Please consider this comment for all the figures in the manuscript.

Answer: All images in this paper was made by author.

Comment 32: References:

Please correct the format for references 4, 6, 162, 222, 227, 228, 232, 243, 244, 263.

Please add journal name, volume, page no. for reference 223.

Answer: The format for references 4, 6, 162, 222, 223, 227, 228, 232, 243, 244, 263 was corrected. References 162, 222, 223, 227, 228, 232, 243, 244, 263 have the new numbers because new information was added.

Comment 33: Comments on the Quality of English Language

The manuscript uses an overly casual tone in a number of places, which compromises the academic language of the manuscript. Further, there is frequent use of uncommon and non-academic keywords (e.g. nanosilver), and typo mistakes are present throughout the manuscript. Thus detailed proofreading is recommended, with an emphasis on using an academic style of writing.

Answer: The detailed proofreading was done for the whole text of the paper. The necessary corrections to improve the text were made.

Reviewer 2 Report

Comments and Suggestions for Authors

A review article entitled Green silver nanoparticles: an antibacterial mechanism by Ekaterina O. Mikhailova presents in detail methods and steps in the synthetic procedures of silver nanoparticles by living organisms, such as bacteria, fungi, plants, etc. Along with a fantastic review of the literature considering the subject, the mechanisms of AgNPs’ antimicrobial activity, together with their anti-biofilm and anti-quorum sensing activities, and toxicity are also very concisely and thoroughly discussed. In my opinion, this review article is great and should be published in the journal Antibiotics, after minor changes regarding some small, technical and grammar errors –

1. All Latin names of bacteria/fungi/algae,… should be in italics,

2. Scanning Electron Microscopy (SEM) is repeated twice in the same sentence as a method (rows 60-61),

3. The period as a punctuation mark was, probably accidentally, placed in several places where it should not be. 

Author Response

Dear colleague,

Thank you very much for your comprehensive analysis of our work and for your kind feedback!

Appropriate changes have been made in accordance with your comments.

Comment 1: All Latin names of bacteria/fungi/algae,… should be in italics.

Answer: All Latin names of organisms were corrected to be in italics.

Comment 2: Scanning Electron Microscopy (SEM) is repeated twice in the same sentence as a method (rows 60-61).

Answer: The unnecessary sentence was deleted.

Comment 3: The period as a punctuation mark was, probably accidentally, placed in several places where it should not be.

Answer: The detailed proofreading was done for the whole text of the paper. The necessary corrections to improve the text were made.

Reviewer 3 Report

Comments and Suggestions for Authors

The manuscript “Green silver nanoparticles: an antibacterial mechanism” has significant potential in the diverse field of pharmaceuticals/biologics to industrial applications. However, the review work needs some major analysis to strengthen and support the review work.

Comments:

1.     In the introduction part, paragraph 1, mentioned “new, highly effective drugs”, Highlighting the example of some effective soluble and insoluble drugs may strengthen this paragraph?

2.      Eco-friendly synthesis using bacteria, fungi, algae and plants all are widely used, anything new?

3.     The introduction part needs more details of the paragraph containing antibacterial properties of AgNPs such as concentration range, size, stability etc., to strengthen and support the review paper.

4.     In the mechanism of synthesis section, you need to provide several types of method and compare your green synthesis method. Why your method is superior than the other technique?

5.     Why your method is nano- synthesis? What would be status/fate of some of the microparticle observed during nano synthesis?

6.     What is the solubility of synthesized silver nanoparticles?

7.     What types of Functional groups contribute for releasing electrons?

8.     Provide the comparable stability study for the silver nanoparticles and its aggregates?

9.     Provide some SEM/TEM images of the silver nanoparticles synthesized from the green sources and demonstrate the morphology of the particles?

10.  Provide a stepwise reaction mechanism of the AgNPs synthesis from any of the sources as an example? Provide a route for the AgNPs biosynthesis?

11.   What would be the optimal concentration range could be use for silver nanoparticles synthesis from fungi?

12.  The fungal mechanism of AgNPs synthesis most likely involves the participation of negatively charged carboxyl groups in proteins? What role does play by carboxyl groups? Make a clear explanation of/with the reference?

13.   Need to provide the references and analysis for the synthesis of different green sources produce silver nanoparticles characteristic observation, such as particle size, stability, zeta potential, functional group, morphology?

14.  Explore some large and small size (kDa) protein contribution for silver nanoparticle synthesis?

15.  What makes Ag nanoparticle a promising nanoparticle? What about other nanoparticle such as gold, palladium, cobalt, zinc? May need to show the comparable toxicity profile of silver nanoparticles and compared the toxicity with other types of metal nanoparticle to support this review work?

16.  Why capping agents? Why not surfactants/ polymer stabilizer?

17.  Provide the surface zeta potential ranges and references of AgNPs with suitable examples/

18.  Provide some of the examples of therapeutic drugs with the biofilms formation? What is the morphology of the biofilm?

19.  How the capping agents interact with the silver nanoparticles stabilization?

20.  Provide a schematic of the in vitro/in vivo toxicity profile of the synthesized green silver nanoparticles?

21.  The conclusion needs to revised and needs to shorten the details?

Author Response

Dear colleague,

Thank you very much for your comprehensive analysis of our work!

Appropriate changes have been made in accordance with your comments. The changes are highlighted in blue.

Comment 1: In the introduction part, paragraph 1, mentioned “new, highly effective drugs”, Highlighting the example of some effective soluble and insoluble drugs may strengthen this paragraph?

Answer: In this sentence only the search for new means of combating bacterial diseases, primarily alternatives to antibiotics, is being discussed. There are many alternatives to antibiotics for the treatment of specific diseases, including therapy with bacteriophages, predatory bacteria, bacteriocins, antimicrobial proteins, plant-derived antimicrobial substances, probiotics, and competitive destruction of pathogens. Commercial examples of bacteriocin solutions are BioSafe™ (bacteriocin: Nisin A), Bactoferm™ F-LC (sakacin A and pediocin PA-1/AcH), ALCMix1 (plantaricin and carnocin), Bactoferm™ (Leucocin or Sakacin), MicroGARD® (a mixture of different bacteriocins). Successful examples of bacteriophage commercialization: Agriphage (Omnilytics Ltd.), Listex (Micreos, Ltd.), SalmFresh®, ListShield®, and EcoShield® (Intralytix Ltd.), show a foreseeable future for the phage therapy. Probiotics are also very popular, for example, ProbioSlim (ProbioSlim, USA, Canada), Activia (Danone, France) etc. However, none of these methods has yet demonstrated effectiveness comparable to antibiotic treatment. The advantage of these approaches is that the treatment is aimed only at the pathogenic bacterium, and not at other members of the commensal, beneficial microbial communities of the host. This is an important difference from most antibiotics, which generally have collateral effects on commensal bacteria in addition to the pathogenic target. It is challenging to discuss solubility in this context, as the spectrum of pharmaceuticals is wide. The pertinent data has been incorporated into the introductory section, with pertinent links provided. The pertinent data was incorporated into the Introduction section, with pertinent references provided.

Comment 2: Eco-friendly synthesis using bacteria, fungi, algae and plants all are widely used, anything new?

Answer: The biological synthesis of nanoparticles (not only silver nanoparticles) by living organisms is an inexhaustible source of medicines that enable the solving of large-scale problems in the treatment of various diseases including bacterial infections, parasitic invasions, inflammatory processes, oncological diseases etc. Currently, there is literature data on the synthesis of AgNPs not only using plant, algae, and lichen extracts, but also using bacterial and fungal cells. Additionally, some information is available about single biological compounds that can conduct this process. This information is provided in the Conclusion section.

Comment 3: The introduction part needs more details of the paragraph containing antibacterial properties of AgNPs such as concentration range, size, stability etc., to strengthen and support the review paper.

Answer: It is difficult to provide brief information about the size, shape, and concentration of green silver nanoparticles as antibacterial agents in this short space. Data on nanoparticle concentration is particularly challenging, as it depends on the biological source used for nanoparticle synthesis. Some information was added to the Introduction section. The relevant references were provided. More detailed data on the size, synthesis conditions, and stability of silver nanoparticles are presented in the Synthesis mechanism section.

Comment 4: In the mechanism of synthesis section, you need to provide several types of method and compare your green synthesis method. Why your method is superior than the other technique?

Answer: Chemical and physical methods were traditionally applied for the synthesis of silver nanoparticles. However, their use is accompanied by several drawbacks. For the physical methods (for example, a laser irradiation method, laser pyrolysis, electrospraying, laser ablation, etc.) require expensive equipment and high energy consumption. On the other hand, the main disadvantages of the chemical methods are supposed to use highly toxic reagents (for example, using sodium borohydride (NaBH4), formaldehyde, methoxypolyethylene glycol, etc.), environmental pollution, carcinogenic solvents, contamination of precursor. An important difference from physical and chemical methods is the direct biocomponent participation (proteins, enzymes, various phytocompounds (flavonoids, terpenoids, tannins etc.)) in the nanoparticle reduction and stabilization. This information was added to the Mechanism of synthesis section. The relevant references were provided.

Comment 5: Why your method is nano- synthesis? What would be status/fate of some of the microparticle observed during nano synthesis?

Answer: A nanoparticle is a nanoobject whose linear dimensions in all three dimensions are in the nanoscale (the linear size range is approximately from 1 to 100 nm). The upper limit of this range is considered to be approximate, because basically the unique properties of nanoobjects do not appear beyond it. Since the size of the biosynthesized silver nanoparticles is in this range, it can be considered nano-synthesis.

Comment 6: What is the solubility of synthesized silver nanoparticles?

Answer: Unfortunately, information on the solubility of green silver nanoparticles has not been found. But we will be very grateful to dear reviewer, if you recommend publications on this issue.

Comment 7: What types of Functional groups contribute for releasing electrons?

Answer: For example, Functional groups like hydroxyl (-OH) and carboxyl (-COOH) in these biomolecules play a crucial role in the reduction process. Upon dissolution in water, AgNO3 dissociates into Ag+ cations and NO3 anions. The negatively charged O⁻ in phenols or COO⁻ in organic acids establishes electrostatic interactions with the positively charged Ag+ ions. This interaction enables the donation of electrons, reducing Ag+ to Ag0 and forming silver nanoparticles (AgNP). The information was added.

Comment 8: Provide the comparable stability study for the silver nanoparticles and its aggregates?

Answer: Unfortunately, it is not entirely clear what was meant by stability. Various biomolecules contribute to nanoparticle stabilization. The protein carboxylate group has a high affinity for acting as a surfactant, forming a protein layer on nanoparticles and ultimately stabilizing AgNPs. Proteins can bind to silver nanoparticles via free amino groups or cysteine residues. Zeta potential value is highly related to stability and dispersion of AgNPs. The supplemental information was added to the section “by plants”.

Comment 9: Provide some SEM/TEM images of the silver nanoparticles synthesized from the green sources and demonstrate the morphology of the particles?

Answer: SEM image of the silver nanoparticles synthesized from the Anadenanthera colubrina extract was added. The relevant reference was provided.

Comment 10: Provide a stepwise reaction mechanism of the AgNPs synthesis from any of the sources as an example? Provide a route for the AgNPs biosynthesis?

Answer: The stepwise synthesis mechanism is shown in Figure 1. Synthesis using various biological sources assumes the presence of all these stages of the process. For example, silver nitrate solution was prepared and added dropwise to leaf extract of Dypsis lutescens (tropical foliage plant) [J. Chil. Chem. Soc., 67, N°2 (2022)]. The aqueous solution was incubated at room temperature for 24 h in dark conditions to prevent photochemical reactions. The obtained solution was kept in an orbital shaker overnight to ensure the synthesis of silver nanoparticles. The hydroxyl groups present in biomolecules are responsible for the bioreduction of Ag+ ions. Flavonoids contain various functional groups, which have an enhanced ability reduce metal ions through the production of reactive hydrogen atom due to tautomeric transitions. During this transition, the enol-from is converted to keto-form, the process ensured by the reduction of metal ions into metal NPs. Proteins have a strong ability to create bonding on metal ions and may be encapsulated around the nanoparticles to avoid agglomeration which makes stabilization in an aqueous solution. During AgNPs formation, both reducing and capping mechanism plays a critical role. Initially, the NPs was confirmed visually through the color changes from light yellow to dark brown which was due to the collective oscillation of free conduction electrons leads to surface plasmon resonance. Another example is for fungal synthesis. The fabrication of silver nanorods by P. sorghina might be due to the secretion of anthraquinones derivatives. Authors have proposed a three-step mechanism for mycosynthesis of silver nanorods. First step is nucleation, which involves the role of proteins acting as capping agent and anthraquinone derivatives to initiate the silver nanorod fabrication. Second step is elongation in which anthraquinone derivative acts as electron shuttle, which takes up the electron donated by inorganic nitrate and photosensitized aromatics compounds from fungal filtrate, by transferring it to silver ions and leading to the reduction of them to form AgNP’s (Ag0). Third and final step is termination of silver nanorod synthesis process, the process will be terminated once the anthraquinone molecule involved in synthesis is either recruited by other nucleation center for the elongation or till the distance an anthraquinone can act as an electron shuttle [Biotechnol Appl Biochem. 2013, 60, 482-93]. This information was added with reference.

More detailed information on the participation of various bio compounds in the synthesis of nanoparticles is presented in the section “The mechanism of synthesis”.

Comment 11: What would be the optimal concentration range could be used for silver nanoparticles synthesis from fungi?

Answer: For instance, P. sorghina showed increase in the mycosynthesis of AgNP’s with increase in the substrate concentration up to 0.9 mM, with the optimum mycosynthesis at 0.9 mM silver nitrate concentration, and the rate of mycosynthesis of AgNP’s by P. sorghina, increased with an increase in fungal filtrate concentration [Biotechnol Appl Biochem. 2013, 60, 482-93]. The information was added to the section “by fungi”.

Comment 12: The fungal mechanism of AgNPs synthesis most likely involves the participation of negatively charged carboxyl groups in proteins? What role does play by carboxyl groups? Make a clear explanation of/with the reference?

Answer: This sentence has been rephrased and clarified with the addition of an appropriate reference. The proteins that can bind to nanoparticles either through free amine groups or cysteine residues in the proteins and through the electrostatic attraction of negatively charged carboxylate groups in enzymes present in the cell wall of mycelia. Another example, is plant biosynthesis. Peptide bonds in proteins and enzymes present in Terminalia arjuna leaf extract are broken down to form smaller peptides. Carboxylate groups in these peptides have a high affinity for the action as surfactants and form a protein corona layer on AgNPs, stabilizing them.

Comment 13: Need to provide the references and analysis for the synthesis of different green sources produce silver nanoparticles characteristic observation, such as particle size, stability, zeta potential, functional group, morphology?

Answer: The information about size, morphology, stability and zeta potential was added to the appropriate section “The mechanism of synthesis”. Additionally, data was shown in the Tables 1, 2.

Comment 14: Explore some large and small size (kDa) protein contribution for silver nanoparticle synthesis?

Answer: 85-kDa protein acts as a capping material and confers stability to the fungal silver nanoparticles. A 59 kDa protein and other proteins (69, 73, 82, 102, and 181 kDa) could participate in the Stereum hirsutum-AgNPs synthesis. Two proteins (36 and 40 kDa) were detected in the extract and AgNPs synthe-sized by Trichoderma harzianum, which were identified as b-1,3-glucanase and chitinase enzymes, respectively. In the synthesis of AgNPs by Aspergillus flavus, a 32 kDa protein, participates in the Ag+ reduction and a 35 kDa acts as a capping agent of nanoparticles. The information was added to the sections “The mechanism of synthesis” and “capping agents”.

Comment 15: What makes Ag nanoparticle a promising nanoparticle? What about other nanoparticle such as gold, palladium, cobalt, zinc? May need to show the comparable toxicity profile of silver nanoparticles and compared the toxicity with other types of metal nanoparticle to support this review work?

Answer: Silver nanoparticles synthesis from biological extract is receiving a greater attention because of their potential application in different fields including medicine. The prospective medical applications (antimicrobial, anticancer treatment, antioxidant, anti-inflammatory activities, wound healing, application in medical equipment, etc.) of AgNPs obtained using a wide variety of biological objects are extremely large, and the number of publications on this topic only continues to increase from year to year. In addition, the bactericidal effect of silver ions, which has been known to mankind for a long time, attracts a significant number of researchers. At the same time, it is challenging to conduct a comparative analysis of the toxicity of silver nanoparticles and other metal nanoparticles, as various biological sources are used for their synthesis and the selection of synthesis conditions involves different temperatures, pH levels, salt concentrations, and other factors. Shape, size and capping agents are also important in this case. Unfortunately, there is very little data on the comparative analysis of the toxicity of nanoparticles of various green nanoparticles. For instance, Panax ginseng-mediated AuNPs were not cytotoxic to HaCaT and 3T3-L1 non-cancerous cells. P. ginseng-generated AgNPs did not exhibit any significant cytotoxic effects on HaCaT cells; however, they showed rather detrimental effects for 3T3-L1 pre-adipocyte cells. AgNPs mediated by P. ginseng were toxic to B16 murine tumor cells, but were comparatively less harmful for human dermal fibroblasts. On the other hand, P. ginseng mediated-AuNPs were non-toxic on either human fibroblast or murine cancer cells [Int. J. Nanomed. 2017;12: 709–723]. Dendropanax morbifera-AgNPs and AuNPs were examined for cytotoxicity in a human keratinocyte cell line and in A549 human lung cancer cell line. The results indicated that AgNPs exhibited less cytotoxicity in the human keratinocyte cell line at 100 µg/mL after 48 hours. On the other hand, AgNPs showed potent cytotoxicity in the lung cancer cells at the same concentration after 48 hours, whereas AuNPs did not exhibit cytotoxicity in both cell lines at the same concentration [Int J Nanomedicine. 2016 Aug 10;11: 3691-701]. In vitro cytotoxic analysis revealed that up to 50 μg/mL-1 concentration AuNPs from rhizome of Anemarrhena asphodeloides did not exhibit any toxicity on 3T3-L1, HT29 and MCF7 cell lines, while being specifically cytotoxic to A549 cell line. On the contrary, Anemarrhena asphodeloides-AgNPs displayed a significantly higher toxicity in comparison to AuNPs in all cell lines specially MCF7 cell line. ROS generation was not affected by AuNPs, on the other hand, AgNPs treatment exhibited a higher potential to induce oxidative stress in A549 cells than HT29 and MCF7 cells [Artif Cells Nanomed Biotechnol. 2018;46(sup2):285-294]. This information was added to the paragraph “in vitro”, section “Toxity” with appropriate references.

Comment 16: Why capping agents? Why not surfactants/ polymer stabilizer?

Answer: Capping agents are of great importance as stabilizers that prevent excessive growth of nanoparticles and prevent their aggregation/coagulation during colloidal synthesis. Different types of capping agents have been used in nanoparticles’ synthesis including surfactants, small ligands, polymers, dendrimers, cyclodextrins, and polysaccharides. In the case when the green nanoparticle biosynthesis was used via extracts of plants, fungi, lichens, algae, bacteria, biological compounds synthesized by these organisms act as capping agents. Proteins, enzymes, exopolysaccharides, flavonoids, tannins, alkaloids, saponins, terpenoids and other biomolecules can be the capping agents.

Comment 17: Provide the surface zeta potential ranges and references of AgNPs with suitable examples.

Answer: The information about zeta potential of green AgNPs was added to the “The mechanism of synthesis” with the references. For example, AgNPs synthesized using the intracellular extracts of B. subtilis and B. megaterium were found to be stable, with zeta potential values of −34.1 mV and −33.9 mV. The zeta potential further confirmed the results of UV-vis spectra. the added Na+ might bind with these biomolecules, which make the AgNPs unstable. The zeta potential values of AgNPs synthesized with E. camaldulensis and T. arjuna were -26 mV ± 4.61 mV and -20 mV ± 5.09 mV, respectively.

Comment 18: Provide some of the examples of therapeutic drugs with the biofilms formation? What is the morphology of the biofilm?

Answer: Combined methods using various antibiotics help in the destruction of biofilms. For example, clarithromycin along with vancomycin was found to destroy the biofilm forming bacterial cells. This combination is active against Gram negative bacteria and specifically proven effective against Ps. aeruginosa and Staphylococcus sp. This combination targets the major component of the EPS matrix i.e., the alginate which is thick and solid difficult to destroy preventing the entry of antibiotics. Another combination of a macrolide and a carbapenem, i.e roxithromycin and imipenem helps white blood cells penetrate inside the matrix and destabilize the biofilm eventually eradicating it. The combination of clarithromycin and levofloxacin with an efficacy rate of 99% was effective against Ps. aeruginosa compared to individual antibiotic therapy. The combination of N-Acetylcysteine NAC (4890ug/ml) and ciprofloxacin (32 or 64ug/ml) had a synergistic effect. This combination also showed antibiofilm activity against Ps. aeruginosa and other microbes, NAC was thought to inhibit EPS matrix production which is one of the significant steps in destroying the rigidity of the biofilm.

Comment 19: How the capping agents interact with the silver nanoparticles stabilization?

Answer: Various biomolecules contribute to nanoparticle stabilization. Enzymes, glycosides, and saponins can help stabilize the nanoparticles. The proteins in biologically synthesized AgNPs solutions contribute to the nanoparticle stabilization. Protein components in the medium can bind to AgNPs through free amino or cysteine groups, acting as capping agents that favour nanoparticle stability in solution and prevent their agglomeration. The complexation of polyphenols with metallic silver and this bond with biomolecules is responsible for the nanoparticle stabilization.

Comment 20: Provide a schematic of the in vitro/in vivo toxicity profile of the synthesized green silver nanoparticles?

Answer: The toxic effects of silver nanoparticles are most often considered in vitro on various human cells. These studies aim to provide a better understanding of the potential risks and benefits associated with the use of AgNPs in medicine. Compared to in vitro studies, much less information is available on the possible mechanisms of AgNP toxicity from in vivo studies. To improve the understanding of the toxicity of silver nanoparticles in vitro and in vivo, the “Toxity” section was divided into appropriate paragraphs.

Comment 21: The conclusion needs to revised and needs to shorten the details?

Answer: The conclusion was shortened in some details.

Round 2

Reviewer 1 Report

Comments and Suggestions for Authors

Thank you for resolving the comments.

Author Response

Dear colleague,

Thank you very much for taking the time to review my manuscript and for your valuable comments!

Reviewer 3 Report

Comments and Suggestions for Authors

After checking the review response, I found satisfactory and enhanced improvement in the draft; I am providing a response for acceptance of this work to this antibiotics journal. 

Author Response

(The authors gave the same response as above.)
